# LiSeCo: Linear Semantic Control for Language Generation

**Emily Cheng**  *emilyshana.cheng@upf.edu*
*Universitat Pompeu Fabra, Barcelona, Spain*

**Carmen Amo Alonso**  *camoalon@stanford.edu*
*Stanford University, CA, USA*

**Reviewed on OpenReview:** *https://openreview.net/forum?id=a3o2pzZuvE*

## Abstract

The prevalence of Large Language Models (LLMs) in critical applications highlights the need for controlled language generation methods that are both computationally efficient and enjoy performance guarantees. To address this need, we use a common model of concept semantics as linearly represented in an LLM's latent space. In particular, we take the view that natural language generation traces a trajectory in this continuous semantic space, realized by the language model's hidden activations. This view permits a control-theoretic treatment of text generation in latent space, in which we propose Linear Semantic Control (LiSeCo), a lightweight, gradient-free intervention that dynamically steers trajectories away from regions corresponding to undesired meanings. In particular, we propose to directly intervene, in an online fashion, the activations of the token that is being generated in embedding space. Crucially, LiSeCo does not simply steer activations towards a desirable region. Instead, it relies on classical techniques from control theory to precisely *control* activations in a context-dependent way, and guarantees that they are brought into a specific pre-defined region of embedding space that corresponds to allowed semantics. The intervention is computed in closed form according to an optimal controller formulation, minimally impacting generation time. This control of the activations in embedding space allows for fine-grained steering of attributes of the generated sequence. We demonstrate that our approach is effective on different tasks—toxicity, sentiment, and language (English/Spanish) steering—while maintaining text quality.

## 1 Introduction

Large Language Models (LLMs) have become widespread in critical applications such as content moderation and real-time information dissemination (Zeng et al., 2024). As a result, strategies that enforce constraints on LLMs' generated text are increasingly needed. To address this challenge, controllable text generation has emerged as a pivotal research area.

Several approaches have been proposed towards controllable text generation (Kumar et al., 2021; Lu et al., 2021; Li et al., 2022; Qin et al., 2022). Of them, a popular approach is prompt engineering (Luo et al., 2023; Bhargava et al., 2023; Cai et al., 2023), where natural language prompts are carefully chosen at input time to steer generation. Other approaches modify LLM weights to achieve the desired outputs (Yao et al., 2023; Li et al., 2023b). Lastly, some methods engineer model *activations* to steer them into the representations of desired outputs (Dathathri et al., 2019; Hernandez et al., 2023; Konen et al., 2024; Li et al., 2024a; Rodriguez et al., 2024; Wu et al., 2024; Im & Li, 2026).

Despite current efforts, ensuring the controllability of these models remains a challenge. In particular, existing methods offer steering capabilities, rather than true *control* over model behavior. Current methods typically nudge a target attribute in a certain direction (Li et al., 2023a; Turner et al., 2023; Rodriguez et al., 2024; Wu et al., 2024), but lack formal guarantees on the steering outcome. For instance, ReFT (Wu

et al., 2024) operates solely on prompt representations and do not intervene in the model's activations as generation unfolds; ActAdd (Turner et al., 2023), like other methods (Li et al., 2023a; Rodriguez et al., 2024), indiscriminately applies a fixed-size intervention without verifying whether the steering goals is already satisfied. Thus, achieving robust and verifiable controllability remains a critical goal for safe deployment of LLMs. We clarify this distinction by defining two terms that are often conflated in the literature: *steering* and *control.*

---

**Steering vs. Control**

**Steering** refers to interventions that bias the model's internal representations or outputs toward a desired outcome—such as reduced toxicity or positive sentiment—without enforcing guarantees on success. Most steering methods operate by learning a direction in latent space that correlates with desired outputs and nudging the model towards it. In contrast, **control** implies an explicit mechanism that enforces constraints on model representations with *formal guarantees.* A controlled generation system *ensures* that representations or outputs lie within a well-defined, often numerically specified, set of acceptable values.

---

Control of model *behavior* is an open problem. However, as a step towards controlling model behavior, we focus on control at the internal *activation* level. To this end, we propose to use control theory to tackle controlled language generation. Specifically, optimal control theory (Kirk, 2004) offers principled methods to steer trajectories in latent space that enjoy theoretical guarantees on the performance of the intervention. Our intervention method, which we call Linear Semantic Control (LiSeCo), derives from a theoretical formulation of controlled text generation. Our contributions are both theoretical and empirical: (1) we formally pose LLM control in activation space as a constrained optimization problem and provide its closed-form solution with guarantees on where the resulting activations lie in that space; (2) we study how control in the activation space can, in theory, translate to controllable token generation during decoding, and (3) we empirically demonstrate our method on text attribute steering for toxicity, sentiment, and output language. We confirm, with experiment corroborating theory, that LiSeCo dynamically controls the activation's trajectory during generation to avoid disallowed concepts while maintaining text quality and minimal impact on inference time latency. Experimentally, we show that principled *control in activation space* lends itself to reliable *steering of the output attributes.*

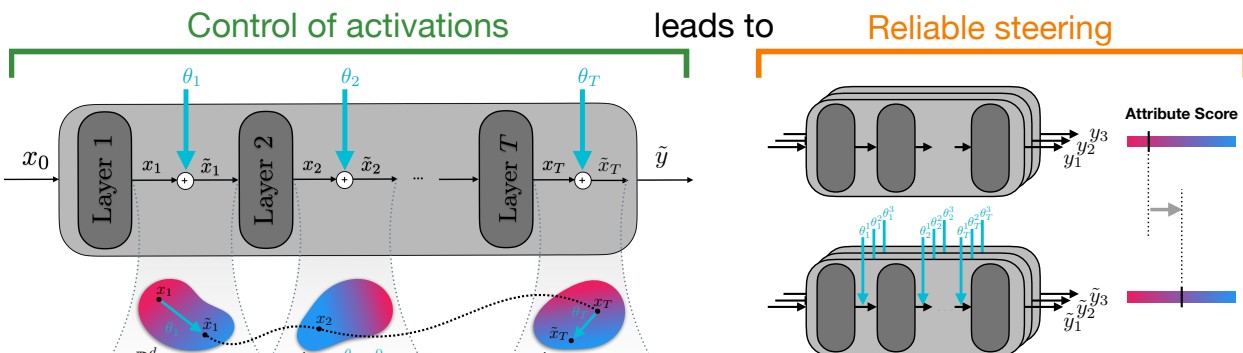

Figure 1: The LiSeCo intervention is computed as the solution to an **optimal control problem**, whose value is dependent on the current activation ($x_t \in \mathbb{R}^d$). When activations naturally fall inside some pre-defined bounds, the value of the intervention is zero. However, when activations fall outside of some pre-defined bounds, the intervention *controls* the activation to *guarantee* that the updated state $x_t + \theta_t \in \mathbb{R}^d$ lies in the desired location of the space. This precise control in the activation space yields to fine-grained steering of the output sequence in token space according to the attribute of interest.

**Contributions of LiSeCo**

LiSeCo, like most steering methods in the literature (Li et al., 2023a; Rimsky et al., 2024; Rodriguez et al., 2024), intervenes in LLM activation space by adding a lightweight vector. However, LiSeCo differs from past approaches in that the latter do not provide *formal* guarantees on the effectiveness of the steering, compliance, or accuracy with which the control goal is achieved, nor at the activation level neither in the resulting token generations. Instead, LiSeCo controls the activations during generation and is *theoretically guaranteed* to steer them into the allowed region. Our work differs from existing literature by the following novel contributions:

1. **Optimal Control Formalization with Theoretical Guarantees.** We frame online LLM intervention as an optimal control problem, which we use to provide an efficient closed-form solution that ensures outputs fall within a target attribute range. Unlike steering methods, which nudge activations toward desired outcomes without guarantees, our method enforces attribute bounds as hard constraints, constituting true control over activations. The optimal formulation ensures that the intervention is minimal, i.e., it has the smallest magnitude required to transport the activation into the desired region of representation space. This avoids over-steering, which preserves the naturalness of the resulting outputs, and is *context-dependent*, i.e., only intervenes when current activations are outside of the allowed region. This is in contrast to other interventions in the literature (see Results, Section 6). Providing guarantees on the output text attribute requires rigid assumptions; we provide sufficient conditions for activation control to translate to output text control in Appendix C.
2. **Interpretability.** Our approach targets of continuous-valued attributes (e.g., toxicity, sentiment) using interpretable numerical scores. Rather than implicitly influencing the model with an intervention that nudges representations in an ad-hoc direction, we directly specify a target range and guarantee compliance in embedding space. This affords control over generation with a greater level of granularity as compared to existing approaches. For instance, our method allows for bidirectional steering, meaning that it can be used to both lower or increase an attribute in a given range. Moreover, the level of guarantee compliance in *token space*, given compliance in activation space, serves as an interpretability tool that tests the functional representation of concepts and their causal relevance in generation.
3. **Closed-form, Low-latency Online Intervention.** The intervention is computed analytically in closed form, avoiding the need for backpropagation or iterative optimization at inference time. This yields minimal computational overhead compared to methods such as FUDGE (Yang & Klein, 2021) or PPLM (Dathathri et al., 2019).

We extend the vision of Dalrymple et al. (2024) by demonstrating a concrete instantiation of guaranteed safe AI principles in a real-world language modeling task. Though Soatto et al. (2023) apply theoretical tools from control to LLM text generation, to the best of our knowledge, our method is the first to propose a control-theoretic intervention whose theoretical guarantees are validated in practice. We analyze other state-of-the-art activation-based methods, such as ReFT or AcT, under a control theoretic lens in Appendix F.

## 2 Related Work

All controlled generation methods aim to modify some text attribute, such as toxicity, while maintaining fluency. Ultimately, all methods work towards this goal by modifying the LLM's final probability distribution, either directly or indirectly. We can situate where different method classes intervene, viewing an LLM as a series of $T$ function compositions corresponding to the $T$ layers, where $s$ is a sequence of tokens:

$$\mathbb{P}_{LM}(\cdot|s_{<i}) = f_T \circ f_{T-1} \cdots \circ f_1(s_{<i}) := \text{LLM}(s_{<i}).$$

**Decoding-based methods** fix the function $\text{LLM} := f_T \circ f_{T-1} \circ \cdots f_1$ and directly edit its output probability distribution $\mathbb{P}_{LM}(\cdot|s_{<i})$ over the next token $s_i$ (Yang & Klein, 2021; Liu et al., 2021; Krause et al., 2021). These methods require access to an external evaluator whose feedback is used to calibrate token probabilities, which can result in high inference latency.

**Prompt engineering** steers the LLM's output by varying the input context $s_{<i}$, keeping the function $\text{LLM} := f_T \circ f_{T-1} \circ \cdots f_1$ fixed (Luo et al., 2023; Bhargava et al., 2023; Cai et al., 2023; Wei et al., 2022;

Li & Liang, 2021). Prompts are often highly task-specific, requiring either manually crafting or ad-hoc computationally-taxing techniques, and success can be brittle to prompt choice (Weber et al., 2023). While the space of natural language prompts is discrete, LLM weights and activations live in continuous high-dimensional space, which is more expressive; then, rather than search over discrete prompts, other approaches that exploit this expressivity directly intervene in the internals of the model.

Of them, **weight-based methods** modify the functions $f_i$ themselves, which permanently constrains the space of final probability distributions $\mathbb{P}_{LM}$. These methods comprise, e.g., reinforcement learning from human feedback (Ouyang et al., 2022), instruction-tuning, parameter-efficient adaptation (Hu et al., 2022), or targeted weight editing (Meng et al., 2022b; Belrose et al., 2023). In such approaches, weights are modified according to the goal of the controlled generation by, for instance, learning the necessary update (De Cao et al., 2021; Mitchell et al., 2021), or localizing and editing target parameters encoding specific knowledge (Dai et al., 2022; Meng et al., 2022a;c; Li et al., 2024b). Pitfalls range from potential inconsistencies and distortions, to the fact that weight-based methods can only correct errors in the LLM's parametric knowledge, but not in-context (Li et al., 2023b).

**Activation editing methods**, such as LiSeCo, fix $LLM := f_T \circ f_{T-1} \circ \cdots f_1$ but intervene at the domain of each $f_i$, where introducing a steering vector modifies the input to $f_i$ (Li et al., 2023a; Turner et al., 2023). These interventions can be seen as restricting the domain of each $f_i$, eventually constraining the space of probability distributions $\mathbb{P}_{LM}$ when composed up through the layers. A key advantage of activation steering is rapid adaptation that can be made *context-dependent*. Often, this is achieved with *linear interventions*, i.e. interventions $\theta_t$ such that layer $t+1$ receives as input $x_t + \theta_t$, where $x_t$ is the output of layer $t$. An initial work in this domain was Plug and Play (Dathathri et al., 2019), where a linear intervention is computed at every layer. The control goal is encoded as the objective function in an optimization that is then solved via back-propagation, adding significant computational overhead at inference time. Subsequent approaches also compute linear modifications to the latent state, but reduce computational overhead, act on only a few layers (Subramani et al., 2022; Konen et al., 2024), pre-compute steering vectors to avoid back-propagation (Turner et al., 2023), or address the issue of computational efficiency at the expense of optimality (the intervention is not formulated as an optimizer) (Li et al., 2024a). Approaches like REMEDI (Hernandez et al., 2023) or ReFT Wu et al. (2024) find optimal interventions to achieve different target outputs, but these are only used to edit representations in the initial forward pass. Lastly, AcT (Rodriguez et al., 2024) learns an optimal transport map between two distributions of outputs (e.g., toxic and nontoxic), and applies this lightweight map online to the representation being generated. None of these existing methods provide a principled *control* strategy—defined here as one that guarantees the activations meet a precise target specification—rather than merely *steering* them in a general direction with the hope of reaching a desired region, often disregarding intermediate regions. In contrast, our method offers control by provably characterizing the distributional structure of the activation space. This, in turn, enables a more fine-grained steering of the token generation, including bidirectional steering along the full spectrum of the attribute to be controlled.

## 3 Problem Statement

We present the problem studied in this paper, framing it as standard optimal control (Kirk, 2004).

### 3.1 Problem Formulation

Given a language model, controlled language generation aims to steer the model's output into a desired one. We study the problem of setting attributes of the model's output text, like sentiment or toxicity, to a certain range. In practice, this range can be quantified via numerical scores, for example, constraining text perplexity to between 0 and 30, text toxicity to a subset of the Likert scale from 1 to 5, or likelihood of having a positive sentiment greater than a probability of 0.75. Formally, an *attribute* is a map $a : \Sigma^* \to \mathcal{A}$ from a language model's output in $\Sigma^*$ (the space of strings) to a numerical score or categorical label in $\mathcal{A}$. In this work, we consider how to steer the output generation of an *already trained model* towards a user-defined desired range $\mathcal{A}^* \subset \mathcal{A}$. Specifically, the requirements for the generated output sequence are twofold: its latent trajectory (a) is *guaranteed* to lie in an "allowed region" that corresponds to $\mathcal{A}^*$ in output attribute space, and (b) stays

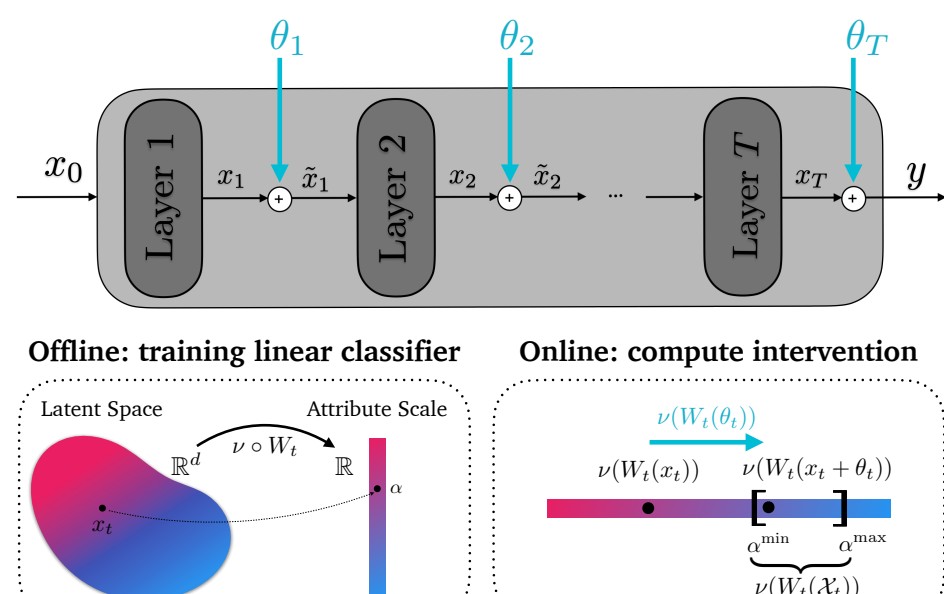

Figure 2: LiSeCo is based on applying a control intervention to the activations after each layer. The intervention is the result of applying a probing classifier $f_t = \nu \circ W_t$, composed of a linear map $W_t$ followed by a nonlinearity $\nu$, mapping from the latent space $\mathbb{R}^d$ to the attribute space $\mathbb{R}$. Given an input sequence, the probe is trained to map the activation of each layer, $x_t$ for every layer $t$, to its corresponding attribute score for the input sequence, $\alpha$. The classifier is then used at inference time to characterize the allowable region $(\mathcal{X}_t)$ to which each latent state $\tilde{x}_t$ is constrained. Keeping trajectories (sequences of $\{x_t\}_{t=1}^T$) out of the disallowed region in latent space is equivalent to keeping their image out of the disallowed region in attribute space. At inference time, the state in latent space $(x_t \in \mathbb{R}^d)$ is mapped via the learned classifier. If it falls outside of the bounds (forbidden region), an intervention $(\theta_t \in \mathbb{R}^d)$ is computed via **optimal control** as to guarantee the updated state $x_t + \theta_t \in \mathbb{R}^d$ lies in the allowable region.

as close as possible to that of the original output sequence, so that text quality is not compromised. In doing this, two questions need to be answered:

1. Given desired attribute scores $\mathcal{A}^* \subset \mathcal{A}$, how can the allowed region be defined for a given language model in latent space?
2. How can an intervention be designed to *guarantee* that the result stays within the allowed region, as defined by scores, while retaining maximal similarity with the original model?

In what follows, we answer the above questions and show that the proposed approach adds minimal computational overhead to language generation without modifying model weights.

## 3.2 Approach

We employ a control-theoretic approach at the level of activations. Given the sequential, feedforward nature of LLM layers, we consider each forward pass to realize a trajectory through the layers' activation spaces. In particular,

$$x_0 = E(s), \quad x_{t+1} = \ell_{t+1}(x_t), \quad y = U(x_T), \qquad \text{with } t = 0, \ldots, T-1 \tag{1}$$

where $E$ and $U$ are the embedding and unembedding maps respectively, $\ell_t$ is the $t^{th}$ LLM layer, $T$ is the number of layers in the LLM, $s \in \Sigma^*$ is the prompt sequence, and $x_t \in \mathbb{R}^d$ is the latent representation of string $s$ after layer $l_t$.

Our strategy is to find a region $\mathcal{X}_t$ of layer $t$'s latent space corresponding to the desired output range $\mathcal{A}^*$. Concretely, we provide a control mechanism by altering each layer's activation, such that latent trajectories

are guaranteed to lie in $\mathcal{X}_t$. That is,

$$x_0 = E(s), \quad \tilde{x}_t = x_t + \theta_t(x_t), \quad x_{t+1} = \ell_{t+1}(\tilde{x}_t), \quad y = U(x_T), \quad \text{with } t = 0, \dots, T-1, \tag{2}$$

where $\theta_t(x_t) \in \mathbb{R}^d$ is a control input to the activation after layer $t$. For each forward pass, this control mechanism is to be applied to a number of layers: as others have shown that semantic steering performs best when done in intermediate layers (Rimsky et al., 2024), the layers to be controlled is a design parameter, $\mathcal{T} \subset [0, \dots, T-1]$, that we explore experimentally in Section 6. We note that this control intervention acts directly on the representation of the token to be generated, is designed online, and it depends on all previous tokens in the sequence.

The goal of this paper is to design the control input $\theta_t : \mathbb{R}^d \to \mathbb{R}^d; x_t \mapsto \theta_t(x_t)$ such that $\tilde{x}_t := x_t + \theta_t(x_t) \in \mathcal{X}_t$ is guaranteed for each intervened layer $t \in \mathcal{T}$. In what follows, we provide an overview of the approach, which we illustrate in Fig. 2 and present in mathematical detail in Section 4.

### 3.2.1 At post-training time (offline): Semantic Probe.

We want to nudge the output attribute towards our desired range $\mathcal{A}^*$ by intervening in latent space. Doing so requires access to the attribute scoring function $a : \Sigma^* \to \mathcal{A}$, which could be given by, e.g., an off-the-shelf neural toxicity scorer. Crucially, we assume the existence of a map $f_t : \mathbb{R}^d \to \mathcal{A}$ that maps the activation space of layer $t$ to output attribute ratings (this assumption is rather strict, see discussion in Appendix C). Let the region $\mathcal{X}_t$ in latent space be the desired output space $\mathcal{A}^*$'s *pre-image* under $f_t$. Then, the problem of constraining $a \in \mathcal{A}^*$ reduces to constraining the activation $x_t \in \mathcal{X}_t$.

To simplify the exposition, let the attribute rating space $\mathcal{A}$ take continuous values in $\mathbb{R}$ (categorical ratings are a straightforward extension of this case). Let $\mathcal{A}^* = [\alpha^{\min}, \alpha^{\max}]$ be the user-defined range of allowed scores. Then for a given layer $t$, the corresponding allowed region $\mathcal{X}_t$ is given by

$$\mathcal{X}_t := \{x \in \mathbb{R}^d \mid \alpha^{\min} \le f_t(x) \le \alpha^{\max}\}. \tag{3}$$

For simplicity, and due to the empirical success of linear probing (Park et al., 2024), we consider the case where $f_t$ consists of a *linear map* followed by a possibly nonlinear *monotonic activation*. That is, at each layer $t$, define $f_t : \mathbb{R}^d \to \mathbb{R}; x_t \mapsto a(s)$, such that

$$f_t(x_t) = \nu(W_t^\mathsf{T} x_t), \tag{4}$$

where $W_t$ is a matrix and $\nu$ a strictly monotonic nonlinear map of the user's choice. This class of functions is expressive enough to handle both linear regression and classification, where in the latter case, the nonlinear activation $\nu$ may be given by the sigmoid to output a probability. Finally, using a training set of labeled examples, which we call the *constraint set*, we learn the linear probe $f_t$ for each layer $t$ via linear regression or classification. This introduces a critical assumption that the target attribute is *linearly represented* in the activation space of each LLM layer (Park et al., 2024), which we verify empirically in Section 6.

We remark that LiSeCo permits an interpretation of activation space in terms of *output constraints* $[\alpha^{\min}, \alpha^{\max}]$, where the output range may correspond to, for instance, human ratings. Importantly, this setup allows us to **directly *control* activations** instead of merely steer them into a particular direction, since we can use the trained probes $f_t$ to restrict attributes to specifically chosen and interpretable scores.

### 3.2.2 At inference time (online): Optimal Control in Semantic Space

In this step, we use the trained probes $f_t$ to intervene the representation at each layer $t$. In particular, we set up the intervention to be *additive*, i.e., the modified representation $\tilde{x}_t = x_t + \theta_t(x_t)$ is the sum of the original representation $x_t$ and the optimal control input $\theta_t(x_t)$, where the intervention $\theta(x_t)$ is computed sequentially at each layer $t$ during the forward pass. In a slight abuse of notation, we abbreviate $\theta_t$ as

$$\theta_t = \theta_t(x_t; W_t, \nu, \alpha^{\min}, \alpha^{\max}) \quad , \text{requiring} \quad \tilde{x}_t = x_t + \theta_t \in \mathcal{X}_t. \tag{5}$$

Mathematically, at layer $t$ we solve an optimization problem over $\theta_t$ where the pre-computed probe $f_t$ enters as a hard constraint in the formulation. This control strategy guarantees that the latent state $x_t$ remains

in the allowed region and retains maximal similarity with the original model. We emphasize that $\theta_t$ is **computed online** and **is gradient-free at inference time**. The specific expression and derivation for $\theta_t$ is provided in Section 4.

## 4 Optimal Controller for Language Generation

In this section, we describe the theoretical contribution of this work. First, we provide the offline and online algorithms for the approach described in the previous section. Then, we provide the mathematical details, expressions, and derivations that ground all of this. In particular, we show the training procedure for the probing classifier that allows for linear (additive) control interventions. Then, using the probing classifier, we design a controller to restrict text generation to the safe region. The optimal intervention is derived in closed form, thus computationally efficient at inference-time. Lastly, we compare existing methods with LiSeCo and prove a control theoretic interpretation for their proposed approaches as well.

### 4.1 Identification of Allowed Region

Given a labeled constraint set $\mathcal{D} = \{s^{(i)}, \alpha^{(i)}\}_{i=1}^N$ of strings $s^{(i)}$ and their attribute scores $a(s^{(i)}) := \alpha^{(i)}$, the first step is to learn a linear probe $f_t$ at each layer $t$. Here, $f_t$ maps the encoded latent state $x_t \in \mathbb{R}^d$ representation of string $s$ to its attribute score $a(s) = \alpha \in \mathcal{A}$.[1]

In the probe training step, the objective is to minimize a layerwise loss with respect to the probe's weights $W_t$

$$\min_{W_t} \frac{1}{N} \sum_{(s,\alpha) \in \mathcal{D}} \mathcal{L}\left(\nu(W_t^\top x_t(s)), \alpha\right), \tag{6}$$

where the function $\mathcal{L}(\cdot, \cdot)$ is a user-defined loss function such as the mean-squared error or cross entropy loss.

Algorithm 1 summarizes the post-training computations to be carried out offline.

---

**Algorithm 1** LiSeCo: Probe-training step (offline)

---
1: **Input:** Labeled dataset $\{(s^{(i)}, \alpha^{(i)})\}$
2: **Output:** Classifier weights $W_t$
3: **for** $t \in \mathcal{T}$ **do**
4:     Extract activations from Eq. 1: $x_t^{(i)} \leftarrow \ell_t(\dots(E(s^{(i)})))$
5:     Train probe using Eq. 6 on $\{(x_t^{(i)}, \alpha^{(i)})\}$ to obtain $W_t$
6: **end for**

---

### 4.2 Optimal Controller Design in Linear Feature Space

We want to design an intervention at each layer $t$ such that the original activation $x_t$ is modified to new one $\tilde{x}_t$, and $\tilde{x}_t$ is guaranteed to lie within the allowed region $\mathcal{X}_t$. Mathematically, we need to derive $\theta_t$ for $\tilde{x}_t := x_t + \theta_t$ such that the goal is satisfied. Here, we show how $\theta_t$ can be seen as the solution to a constrained optimal control problem. We first pose the problem mathematically, and then introduce a relaxation that allows for an efficient online computation of the intervention $\theta_t$.

#### Optimal Control Setup

The optimal controller aims to keep latent trajectories out of the unsafe region without compromising output quality. That is, we perform constrained optimization where latent trajectories maximally approximate the original ones (proxying text quality) while avoiding the unsafe region as defined by the probe. This gives rise

---
[1]Note that the learning task is regression-like, not classification-like—we want the probes to be calibrated to the scoring function, not just the binary labels.

to the following optimization problem:

$$\min_{\{\theta_t\}_{t\in\mathcal{T}}} \quad \sum_{t\in\mathcal{T}} \|\theta_t\|_2^2 \tag{7a}$$

$$s.t. \quad \alpha^{\min} \leq \nu(W_t^\top(x_t+\theta_t)) \leq \alpha^{\max}, \quad \forall t \in \mathcal{T} \tag{7b}$$

$$x_{t+1} = \ell_t(x_t+\theta_t), \quad \forall t = 1,\ldots,T \tag{7c}$$

$$x_0 = E(s), \tag{7d}$$

where $s$ is the prompt sequence string. Optimization problem 7 aims to find the minimum $l_2$-norm intervention $\theta_t$ for $t \in \mathcal{T}$ (Eq. 7a) that satisfies the following constraints: Eq. 7b requires the modified activation $x_t + \theta_t$ be classified as disallowed by the probe $f_t$; Eq. 7c captures LLM dynamics, i.e., layer $t$ maps the modified activation $x_t, \theta_t$ to the next latent state $x_{t+1}$; Eq. 7d states that the LLM's input embeds the input context, so that interventions are *context-dependent*. The intervention that solves optimization problem 7 is *guaranteed by construction* to keep intervened activations $\tilde{x}_t \; \forall t \in \mathcal{T}$ in the allowed region.

Whether attribute control is expressed as a cost or a constraint depends on the use case. Other approaches, in contrast to ours, encode attribute control in the optimization objective, but not via hard constraints (Dathathri et al., 2019; Hernandez et al., 2023). LiSeCo's constrained optimization framework also permits this interpretation by relaxation of constraints; though we leave its testing to future work, we state its equivalent problem and prove its *closed-form optimal solution*, which has only been empirically approximated by hyperparameter search in the literature (Li et al., 2023a), in Appendix M.

**Optimal Controller Computation**

Optimization problem 7 is a standard problem in the optimal control literature, where by Bellman's Optimality Principle, the standard approach to solving it is dynamic programming (Kirk, 2004). That is, the optimal solution is computed for the last layer $T$, then via backward induction for $T-1,\ldots,1$. But, layer dynamics 7c are highly non-convex, and solutions incomputable in closed form, hence their optimality is not guaranteed. Further, dynamic programming requires gradient backpropagation at each LLM forward pass, adding significant inference latency.

To overcome these limitations, we relax problem 7. No longer searching for a globally optimal solution across layers, we now search for locally optimal solutions at each layer. Now, Eqs. 7c and 7d cease to play a role, as each layer is optimized for separately. Then, problem 7 is relaxed such that at each controlled layer $t \in \mathcal{T}$, the intervention $\theta_t$ is defined as the solution to the following optimization problem:

$$\theta_t^* = \operatorname*{argmin}_{\theta_t \in \mathbb{R}^d} \quad \|\theta_t\|_2^2 \tag{8a}$$

$$s.t. \quad \alpha^{\min} \leq \nu(W_t^\top(x_t+\theta_t)) \leq \alpha^{\max} \tag{8b}$$

The sequence of $\theta_t$ that solve problem 8 may not optimize the original formulation 7. However, one is not anyway guaranteed to find global optima due to the high nonconvexity of layer computations. Furthermore, optimality is not essential as the cost aims only to preserve similarity with the original model. Meanwhile, the guarantee to avoid unsafe region $\mathcal{X}_t$ is still enforced via Eq. 8b.

A key advantage of relaxed formulation 8 is that it is solvable in closed-form, per-layer, with minimal computational overhead. The following theorem states the analytical solution for optimal $\theta_t$.

**Theorem 4.1** (Optimal $\theta$). *The optimal solution $\theta_t^* \in \mathbb{R}^d$ to the optimization problem 8 is given by Table 1.*

| **Condition** | $\nu(W_t^\top x_t) > \alpha^{\max}$ | $\nu(W_t^\top x_t) < \alpha^{\min}$ | otherwise |
|---|---|---|---|
| $\boldsymbol{\theta_t^*}$ | $\dfrac{\nu^{-1}(\alpha^{\max}) - W_t^\top x_t}{\|W_t\|_2^2} W_t$ | $\dfrac{\nu^{-1}(\alpha^{\min}) - W_t^\top x_t}{\|W_t\|_2^2} W_t$ | 0 |

Table 1: Optimal value of intervention $\theta_t^*$ at layer $t$.

*Proof.* Proof relies on leveraging the KKT conditions. See Appendix D for details. □

Geometrically, the optimal solution is the vector from $x_t$ to the closest point in $\mathcal{X}_t$. When $x_t \in \mathcal{X}_t$ already, no update is needed; hence $\theta_t^* = 0$. Otherwise, the update is a factor of $W_t$. We note that the value of the *control* intervention $\theta_t^*$ depends on the current latent state $x_t$, and its magnitude and direction are explicitly dependent on $x_t$. This is in contrast to many steering methods, where the activations are often over- and under-steered towards a constant direction with a constant magnitude (Turner et al., 2023; Li et al., 2023a; Rodriguez et al., 2024). Moreover, since $\theta_t^*$ exists in closed-form, computing an intervention incurs negligible computational cost. Crucially, it is guaranteed to keep the latent state outside the disallowed region. Algorithm 2 below summarizes the inference-time usage of LiSeCo.

---

**Algorithm 2** LiSeCo: inference-time deployment (online)

---

1: **Input:** Prompt $s$, control layers $\mathcal{T}$, parameters $W_t$, $\nu$, $\alpha^{\min}$, $\alpha^{\max}$
2: **Output:** Generated token $\tau$
3: $x_0 \leftarrow E(s)$
4: **for** $t \in [1, \ldots, T]$ **do**
5:     Compute activation from Eq. 1: $x_t \leftarrow \ell_t(x_{t-1})$
6:     **if** $t \in \mathcal{T}$ **then**
7:         Compute score from Eq. 4: $\alpha \leftarrow \nu(W_t^\mathsf{T} x_t)$
8:         Solve for $\theta_t$ (from Table 1) using $W_t$, $\alpha^{\min}$, $\alpha^{\max}$:
9:         **if** $\alpha > \alpha^{\max}$ **then**
10:             $\theta_t \leftarrow \frac{\nu^{-1}(\alpha^{\max}) - W_t^\top x_t}{\|W_t\|_2^2} W_t$
11:         **else if** $\alpha < \alpha^{\min}$ **then**
12:             $\theta_t \leftarrow \frac{\nu^{-1}(\alpha^{\min}) - W_t^\top x_t}{\|W_t\|_2^2} W_t$
13:         **else**
14:             $\theta_t \leftarrow 0$
15:         **end if**
16:         Compute modified representation: $x_t \leftarrow x_t + \theta_t$
17:     **end if**
18: **end for**
19: $\tau \leftarrow U(x_T)$

---

**Example: Logistic regression as a case of Theorem 4.1** One can generally adapt Theorem 4.1 to any invertible nonlinearity $\nu$ and bounds $\alpha^{\max}, \alpha^{\min}$ according to the specific use case. One such special case is *logistic regression*, where the nonlinearity $\nu$ is the sigmoid $\sigma$ and we set the likelihood of an attribute above or below a threshold $p \in [0, 1]$. Then, optimization problem 8 can be realized by

$$\min_{\theta_t} \quad \|\theta_t\|_2^2 \tag{9a}$$

$$s.t. \quad \sigma(W_t^\top(x_t + \theta_t)) - p \le 0, \tag{9b}$$

for each layer $t = 1 \cdots T$. Optimal $\theta_t$ is then given by the following corollary:

**Corollary 4.2** (Optimal $\theta$, threshold)**.** *The optimal solution $\theta_t^* \in \mathbb{R}^d$ to the optimization problem 9 is given by*

$$\theta_t^* = \frac{\ln \frac{p}{1-p} - W_t^\top x_t}{\|W_t\|_2^2} W_t \tag{10}$$

*if $\sigma(W_t^\top x_t) > p$, and $\theta_t^* = 0$ otherwise.*

The corollary follows from simply substituting the nonlinearity $\nu$ and bounds $\alpha^{\min}$ and $\alpha^{\max}$ in Theorem 4.1; Theorem 4.1 can be adapted to arbitrary invertible nonlinearities and bounds according to the user's need.

*Remark* 4.3*.* The degenerate case where $\alpha^{\min} = \alpha^{\max} =: p$ corresponds to setting an attribute to a specific value. Although the theory permits this, in practice it is impossible to guarantee that the scores of the

generations will be equal to $p$ due to uncertainty introduced by the classifier or numerical errors. Further robustness analysis to ensure that score is within a ball around $p$ is left for future work.

Finally, although control occurs *locally* at each layer, the local control steps compose to modify the distribution over the next token. To see this, consider a single token generation. Each sequential control action at layer $t$ guarantees that latent state $\tilde{x}_t \in \mathcal{X}_t$ is classified as "allowed", or equivalently, eliminates the set of disallowed trajectories. By the time we reach the last layer $T$, the latent trajectory is guaranteed to have been rated as "allowed" at every preceding intervened layer. Then, we hypothesize that as a result of the control in the activations, the LLM's output is guided towards scoring in the allowable range $\mathcal{A}^*$. This hypothesis is empirically verified in the following experimental sections.

## 5 Experimental Methods

We tested LiSeCo on three separate steering tasks: **toxicity**, **sentiment**, and **output language (English → Spanish)**.

**Models**   We used three state-of-the-art causal language models: Llama-3-8B (Meta, 2024), Gemma-2-2b (Team et al., 2024), and Mistral-7B (Jiang et al., 2023). While the architectural details of a layer (attention + MLP) differ slightly between models, our intervention treats layers as black boxes and operates at the level of the *residual stream* (Elhage et al., 2021). This permits our intervention to be applied as a lightweight layer wrapper, in an architecture-agnostic way.

**Attribute scoring functions**   Recall that LiSeCo is trained with respect to a scoring function $a : \Sigma^* \to \mathbb{R}$ that the practitioner has access to. Therefore, in evaluating whether guarantees hold, we use $a$ to not only label the training points in the constraint set, but also evaluate the generations at test time. While one can use *any* scoring function $a : \Sigma^* \to \mathbb{R}$, we use off-the-shelf neural classifiers from Huggingface. In particular, to score **toxicity**, we choose $a$ to be the RoBERTa-based toxicity scorer (Logacheva et al., 2022) that maps sentences to a likelihood of being toxic in $[0, 1]$. Logacheva et al. (2022)'s classifier is trained on binary classification on Kaggle's Jigsaw dataset (Adams et al., 2017). To score **sentiment**, we similarly choose $a$ to be a RoBERTa-based sentiment classifier (Camacho-collados et al., 2022), trained on annotated Twitter data (Barbieri et al., 2020), which assigns sentences to the likelihood of being negative in $[0, 1]$. To score the **language** generation task, we used the RoBERTa-based language classifier of Conneau et al. (2020), which maps text to the likelihood of being one of twenty languages (including our targets English and Spanish).

### 5.1 Offline step: Probe calibration

**Probe-training dataset**   We test our method on fine-grained steering of text toxicity and negativity. Borrowing terminology from Ashok & Poczos (2024), we first learn probing classifiers $f$ using a labelled *constraint dataset*. Then, we evaluate text generation on a *task dataset*.

For **toxicity**, we use Kaggle's Jigsaw dataset (Adams et al., 2017) as the constraint dataset. The dataset contains 30k label-balanced natural language comments. Then, we use our toxicity scorer to label all sentences in the constraint set to produce a probe training set of (sentence, score) pairs.

As **sentiment** datasets tend to be domain-specific (e.g., movie reviews), we combine several datasets to form the constraint dataset ($N =$ 30k). This consists of +/- label-balanced samples of 7500 datapoints each from IMDb film reviews (Maas et al., 2011), Tweets (Barbieri et al., 2020), Yelp reviews (Zhang et al., 2015), and Amazon reviews (Hou et al., 2024). For preprocessing details, see Appendix G. We score all texts using Camacho-collados et al. (2022) to produce the probe training set of (sentence, score) pairs.

For **language generation (English→Spanish)**, we used the FLORES-Plus dataset (NLLB Team et al., 2024). This contains $N = 2010$ parallel English and Spanish sentences collected from Wikinews, Wikivoyage, and Wikibooks. Because sentences contain only one language, we do not create labels by scoring the texts ourselves. Instead, we simply take the gold binary label (Spanish or English).

**Probing classifiers**   Our theoretical guarantees rely on a key assumption: that at each layer $t$, there indeed exists a $\mathcal{R}_t$ separable by linear $f_t$ which together capture a semantics of the text being generated. We first verify, using a linear probe, that it is possible to learn the text attribute score from each layer of the LLM. Towards this aim, we split each of the constraint datasets into an 80% training set and 20% validation set. Then, for each model, dataset, and layer, we extract the last token hidden representations $x_t \in \mathbb{R}^d$ for each training sequence; we choose the last token embedding to represent the entire sequence, as in causal LLMs, it is the only to attend to the entire input sequence. We then train one binary classifier $f_t$ per-layer to minimize the cross-entropy loss between the probe prediction and ground-truth scorer in $[0, 1]$. See Appendix H for implementation details.

## 5.2   Online step: Text generation

For each LLM, we insert trained probes $f_t$ at each layer to evaluate layer-wise toxicity likelihood at each forward pass. If layer $t$'s representation $x_t$ is evaluated toxic, then the control input $\theta_t$ is dynamically applied. We fix text generation for all methods to max 100 new tokens with top-$p = 0.3$ sampling, a temperature of 1.0 and repetition penalty of 1.2, the same as in published baselines (Rodriguez et al., 2024; Li et al., 2023a).

### 5.2.1   Baselines

To the best of our knowledge, there are no baselines in the literature offering native guarantees. Therefore, we report only on LiSeCo for fine-grained *activation control*, but we compare LiSeCo against existing methods for *attribute reduction* or *induction* on the text generation. For **toxicity and negativity reduction and English → Spanish**, we test several baselines: no-control and prompting with instruction-tuned models, as well as two activation steering methods Activation Addition (ActAdd) (Turner et al., 2023) and Linear AcT (Rodriguez et al., 2024).

**Instruction-tuned models**   All tested models have instruction-tuned variants. During evaluation, we prompt the instruction-tuned model using a template whose instructions are slightly modified from Mistral's system prompt provided in Jiang et al. (2023) (see Appendix I for details).

**ActAdd**   Like LiSeCo, ActAdd steers text generation in activation space (Turner et al., 2023). For each model, the steering vector is computed as follows: (1) a source and target prompt, e.g., ("hate"→"love"), are each fed through the model and activations collected; (2) for each layer, the steering variable is computed as the difference from source to target activation; (3) at inference time, the steering variable is added to the intermediate representations of the input data. Like LiSeCo, ActAdd is gradient-free at inference-time. But, there are key differences: since steers derive from natural language prompts, ActAdd does not require a supervised learning phase on annotated data as in LiSeCo. For the same reason, the method lacks guarantees. For implementation details, see Appendix J.

**AcT**   Similar to LiSeCo, AcT also steers text generation in activation space (Rodriguez et al., 2024). Using an optimal transport framework, an optimal transport map between two distributions of outputs (toxic and nontoxic) is learned offline at post-training. At inference time, this lightweight map is applied online to the activations being generated. Similar to LiSeCo, it is gradient-free at inference-time. However, there are some fundamental differences. In AcT, steering is done in-distribution and, although it can be tuned with a strength parameter, gives coarse control over how much to shift. Moreover, steering is only one-direction (from toxic to non-toxic) and is not used in a bi-directional manner. Moreover, it lacks guarantees on the effect of the interventions on the controllability of the method.

### 5.2.2   Evaluation

We evaluate LLM generations on toxicity and sentiment steering. At the same time, we want our intervention to minimally compromise language modeling performance. To do so, we score generations' toxicity and sentiment, as well as proxy their naturalness using sequence perplexities.

**Test set**  For the inference-time test set, we repurpose the same datasets as the training step. To make the test dataset for each task, we first sampled $N = 1000$ sentences from the respective dataset and truncate each to the first 10 words. For the sentiment and toxicity tasks, the 10 word prompts themselves already contained negative or toxic content. Therefore, in order to create the test set, we performed a further filtering step: for each LLM and 10-word prompt, we first sampled a baseline generation, retaining the set of all prompts that are themselves non-toxic (non-negative), for which the baseline model produced a toxic (negative) *continuation.* Finally, we sampled an equal number of non-toxic (non-negative) prompts that produced a non-toxic (non-negative) continuation. For sentiment and toxicity, this resulted in a balanced test set of equal numbers of "would-be toxic/negative" continuations and "would-be non-toxic/non-negative" continuations. For the sentiment task, this resulted in $N = 500$ balanced prompts per model. For the toxicity task, because models were empirically biased to be non-toxic, we were able to sample fewer samples: $N = 186$ (Llama), $N = 160$ (Gemma), and $N = 180$ (Mistral). For the language task, because baseline models conditioned on an English prompt seldom produce a Spanish continuation and vice versa, we simply subsampled $N = 200$ balanced prompts, 100 English and 100 Spanish. Lastly, during test time, we collect the (intervened) models' continuations, capped at maximum 100 new tokens, for evaluation.

**Semantic control**  We rate text generation toxicity, sentiment, or language (English or Spanish) using the previously described attribute scoring functions. We convert the scorer's ratings into labels, where sequences are labeled toxic (negative, English) if the classifier returns a likelihood higher than 0.5, and non-toxic (positive, Spanish) otherwise. The trained linear probes also provide attribute likelihoods, which we use to post-hoc validate LiSeCo, but not to evaluate text generation attributes per-se. The probe score returns the likelihood that a sequence is toxic (negative, English) as determined by the probes' learned semantics, and is used to evaluate control in activation space.

**Text naturalness**  The applied intervention ideally should not compromise language modeling performance. We quantify performance using the average perplexity (PPL) of generations under a different LLM, Qwen-2.5-3B (Bai et al., 2023). We used a different model family to score PPL, given evidence that LLMs are biased towards their own generations (Long et al., 2024).

## 6 Experimental Results

We first observe that toxicity and sentiment are approximately linearly represented in latent space (Park et al., 2024). We then demonstrate that LiSeCo predictably reduces the controlled attribute as a function of $p$ while maintaining text naturalness. Second, we demonstrate that LiSeCo achieves precise control of activations, such that specifying the desired range $[\alpha^{\min}, \alpha^{\max}]$ indeed controls the activations' probe scores to that range. Finally, we show that LiSeCo performs competitively with existing baselines for attribute reduction while achieving the best naturalness, without extensive finetuning nor online inference latency.

### 6.1 Attributes are approximately linearly represented in latent space

Figure 3 shows, for all models, linear probe validation accuracies per-layer, averaged across 5 random seeds. Probes attain high accuracies of ∼90% for toxicity (Figure 3 left) and ∼ 80% for sentiment (Figure 3 right), confirming the disallowed toxic (negative, English) regions $\mathcal{R}_t$ are approximately linearly decodable, as predicted by the *Linear Representation Hypothesis* Park et al. (2023).

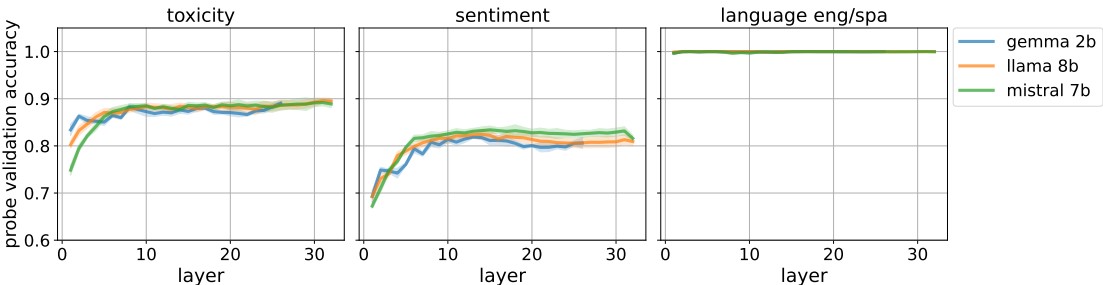

Figure 3: **Linear probe validation accuracy for toxicity (left), sentiment (middle), and language (right) detection.** All curves are shown $\pm$ 1 SD across 5 random seeds. Tasks converge to reasonable accuracies of $\geq 75\%$ for most layers of all models, with mid-layers attaining $\approx 90\%$ for toxicity, above $80\%$ for sentiment, and nearly $100\%$ for language detection.

While we use $80\%$ of the constraint set to train the probes ($N$=24k for toxicity and sentiment, $N$=1.6k for language), we demonstrate in Appendix H that toxicity and sentiment probes can be learned with much fewer samples. Moving forward, toxicity and sentiment results are shown on the original $N = 24$k training set, as training each layer only took 2 minutes on an A30 GPU.

Finally, Figure 3, in line with prior work (Rimsky et al., 2024; Cheng et al., 2025; Lad et al., 2025), suggests to control activations starting from *intermediate layers*, as this is where high-level semantic attributes like sentiment are most linearly decodable. We therefore apply LiSeCo on all layers after layer 8, where probing validation accuracy appears to plateau in Figure 3. This choice is supported by ablation experiments in Appendix M that show multi-layer steering to outperform single-layer steering, and suggest to avoid steering the first few layers of the model.

## 6.2 LiSeCo achieves control with guarantees in activation space

LiSeCo controls activations to the correct safe set. To demonstrate this, we ran LiSeCo for various ranges $[\alpha^{\min}, \alpha^{\max}]$ from $0.01 \pm 0.01$ to $0.99 \pm 0.01$. If LiSeCo truly achieves control in activation space, then we expect the trained probes to score the layer activations, post-intervention, to between $[\alpha^{\min}, \alpha^{\max}]$.

Figure 4 demonstrates this in practice. Figure 4 shows the distribution of the intervened activations' attribute (toxicity, sentiment) scores, scored by the trained probes. Each point is the trained probe's score of a single layer; the bottom row of each plot depicts the activations' score distribution before LiSeCo (brown points), and other rows depict the distribution after LiSeCo. The desired regions of activation space, corresponding to attribute scores $[\alpha^{\min}, \alpha^{\max}]$ computed by the trained linear probes, are shown in green. No matter the LLM or task, LiSeCo systematically controls *activations* (colored points) to the desired range.

## 6.3 Control in activation space translates to reliable steering in output space

Here, we study how control in activation space leads to reliable steering in output space. Specifically, we show that LiSeCo outperforms existing baselines on attribute steering and text naturalness. We show that controlling activations reliably steers the output attribute, where the dependence between LiSeCo $\alpha$ and the output attribute is empirically monotonic, but not identity. Finally, we discuss a path forward for guarantees in activation space to translate to guarantees on the output.

**LiSeCo is competitive with baselines for steering and text quality** Recall that the test set consists $50\%$ generations for which no-control produced a toxic (negative, English) continuation, and $50\%$ for which it produced a nontoxic (nonnegative, Spanish) continuation. We first consider would-be toxic (negative/English) generations. Figure 5 shows, for sentiment (top) and toxicity (bottom), the *safety-naturalness plane*, where the output text attribute is plotted against its perplexity. Each baseline's (safety, naturalness) distribution is

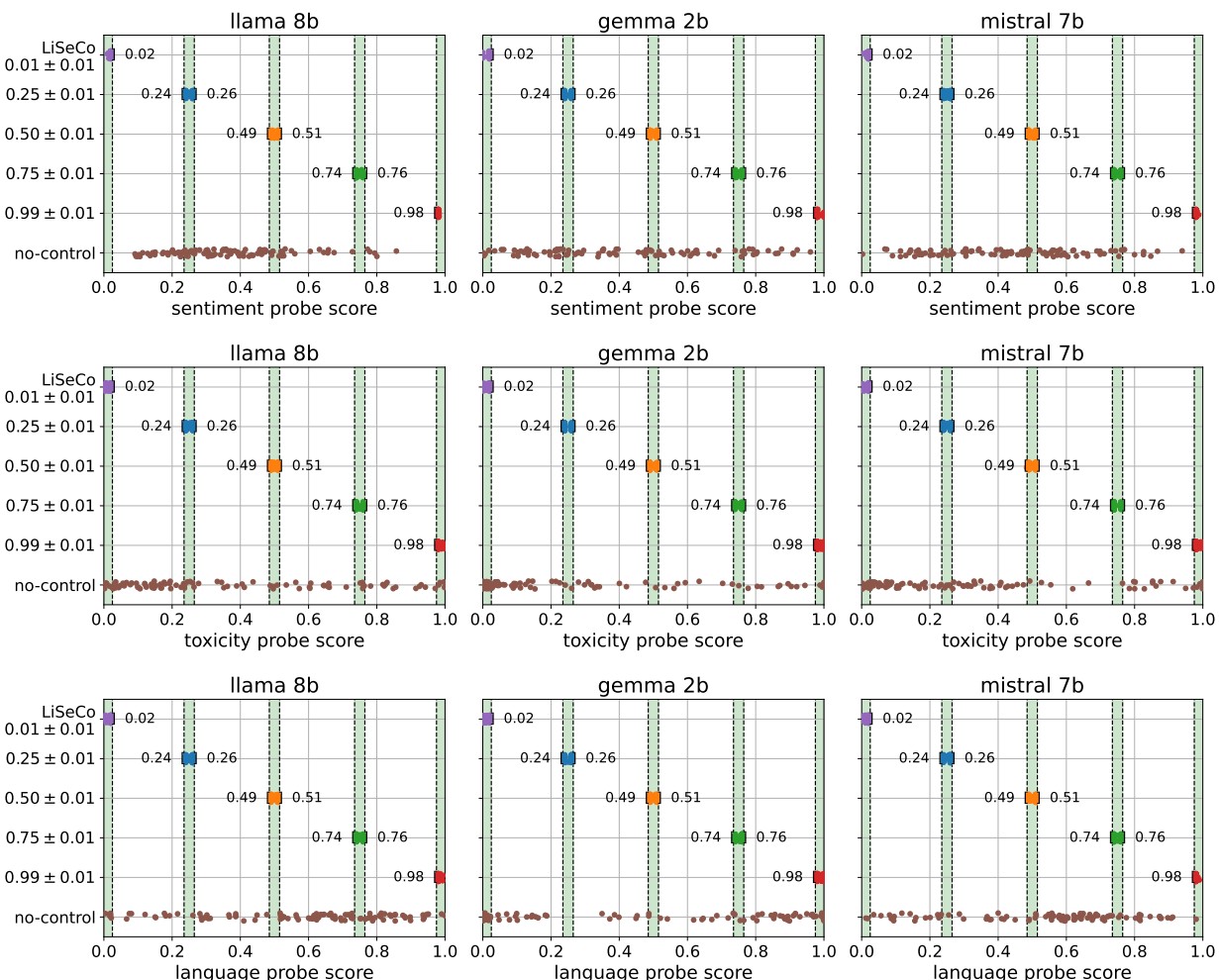

Figure 4: **LiSeCo controls attributes in activation space.** Attribute probe scores for LiSeCo are shown for sentiment (top), toxicity (middle), and language (bottom) on the models Llama, Gemma, and Mistral (left to right). The y-axis of each plot shows the desired range $[\alpha^{\min}, \alpha^{\max}]$, also shaded in green on the plot. The colored points are the actual distribution of the attribute as measured by the trained probes, after applying LiSeCo. A no-control baseline is shown on the bottom row, indicating the distribution if LiSeCo is not applied. In all cases, LiSeCo successfully controls activations to the desired range, seen by the colored dots falling within the green intervals.

shown as an ellipse centered at the mean, shown with one standard deviation. Only the best hyperparameter settings for each baseline are shown (LiSeCo $[\alpha^{\min}, \alpha^{\max}] = [0, 0.005]$.

LiSeCo reduces the desired attribute without sacrificing text naturalness, seen by blue ellipses (LiSeCo) being vertically aligned with the red ones (baseline). For English to Spanish steering (bottom), LiSeCo outperforms baselines by achieving near-perfect performance, while maintaining the lowest perplexity. Interestingly, for English to Spanish steering, where it is simple to tell when the model switches language (as opposed to toxicity/sentiment), the switch tended to be immediate after the prompt in Llama and Mistral, but required several token generations in the case of Gemma, see Appendix N. In the case of negativity reduction (top row), LiSeCo is competitive with AcT (Rodriguez et al., 2024), outperforming all other baselines for both attribute steering and text naturalness. LiSeCo's minimal effect on text naturalness is baked into its design, as it introduces the *minimum norm* intervention when the activation falls into the unsafe region, and does not intervene if the activation is already classified safe. The design choice of minimal intervention, by contrast,

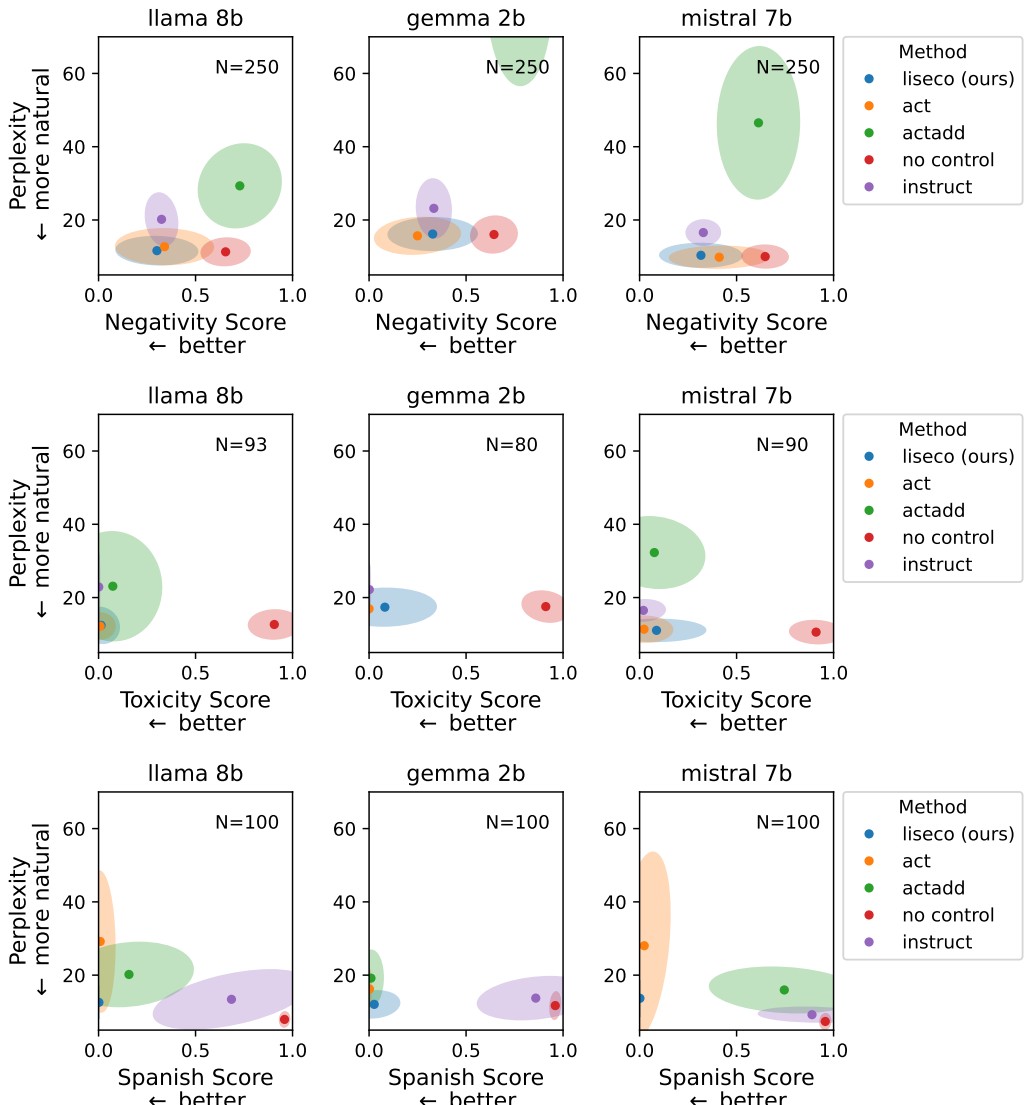

Figure 5: Generations are plotted on the *safety-naturalness plane* (bottom left is best). Each ellipse represents the mean (·) with one standard deviation of a single method; only the best hyperparameter setting is shown for each method. The shaded red region represents the region of the safety-naturalness plane that would be classified negative or toxic by the neural scorer. In general across models and tasks, **LiSeCo performs competitively to baselines while consistently demonstrating high naturalness**. In the negativity reduction task (top row), LiSeCo also demonstrates the best negativity reduction.

is not a part of other baselines, where the intervention is always applied (Turner et al., 2023). To that end, we show in Figure K.1 that LiSeCo *abstains* from intervening if the generation is already classified safe, which maintains a comparable perplexity to the baseline. In practice, LiSeCo preserves the original safety and naturalness distribution as desired, while other methods may negatively impact either factor (see Appendix K).

**Control in activation space permits reliable steering in output space**   We have shown that LiSeCo achieves control in activation space. But, do guarantees in activation space translate to guarantees in generation space? Recall that a key assumption of LiSeCo is that there exists a map $f_t : \mathbb{R}^d \to \mathcal{A}$, where $f_t$ is

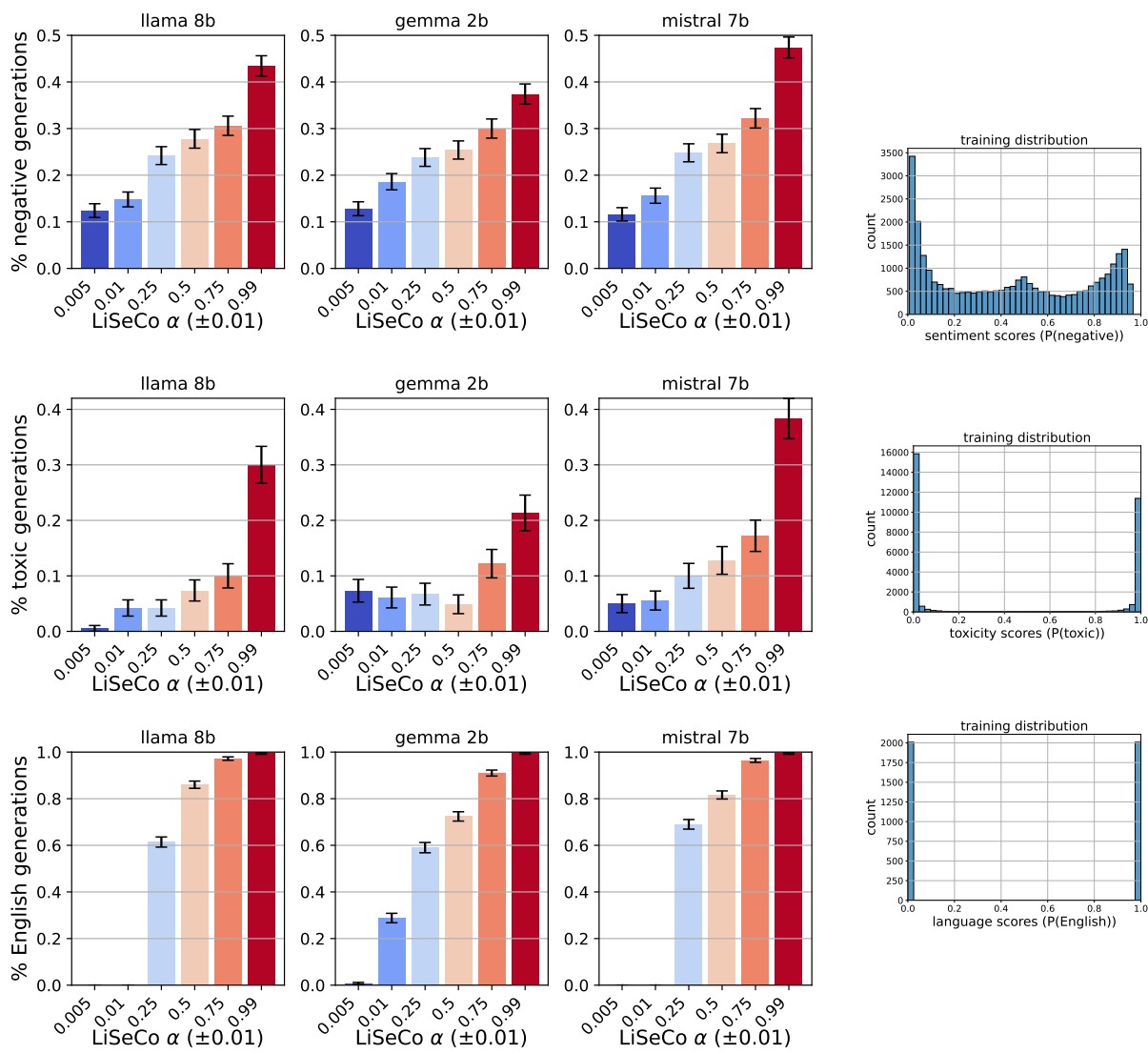

Figure 6: **(Left)** LiSeCo $\alpha$ steers output sentiment (top), toxicity (middle), and language (bottom). We show, for a range of $[\alpha^{\min}, \alpha^{\max}] = \alpha \pm 0.01$ (x-axis), the true proportion of toxic/negative generations (y-axis) with one standard error. For all settings, there is a **clear monotonic trend where smaller LiSeCo $\alpha$ translates to fewer unsafe generations**. Intervention with LiSeCo $\alpha \approx 0.01$, $\alpha \approx 0.99$ significantly changes the distribution of unsafe generations, seen by non-overlapping error bars between gray and dark blue, dark red, respectively. **(Right)** The training set y-label distributions for sentiment scores (top), toxicity scores (middle), and language labels (bottom). Sentiment (top) achieved steerability to a wider range than toxicity (middle), though English→Spanish achieved the entire range of values between 0 and 1.

a linear probe. In order for control in activation space to translate to control in attribute space, Lemma C.1 in Appendix C states that probe $f_t$ needs to apply to *every reachable point* in $\mathbb{R}^d$. Moreover, practically, $f_t$ would need to be perfectly learned from data. These strong assumptions that rarely holds in practice—we explicitly test it on out-of-distribution data in Section 6.4, finding mixed results. Fortunately, empirically on in-distribution data, control in activation space does buy us reliable *steering* in output space.

Figure 6 shows a clear monotonic trend between LiSeCo $\alpha$ and the number of unsafe generations. For all models and tasks, smaller LiSeCo $\alpha$ (x-axis) predictably decreases the proportion of toxic/negative generations (y-axis). This allows LiSeCo $\alpha$ to act as an interpretable knob that one can adjust at inference-time to obtain

|                 | llama 8b | gemma 2b | mistral 7b |
|-----------------|----------|----------|------------|
| in-distribution | $0.987_{0.0002}$ | $0.913_{0.0112}$ | $0.988_{0.0002}$ |
| poems           | $0.987_{0.0016}$ | $0.816_{0.0349}$ | $0.989_{0.0003}$ |
| europarl        | $0.986_{0.0007}$ | $0.607_{0.0461}$ | $0.986_{0.0005}$ |
| code            | $0.889_{0.0282}$ | $0.287_{0.0445}$ | $0.931_{0.0220}$ |

Table 2: **English $\rightarrow$ Spanish OOD Results, Average Prob(Spanish) $\uparrow$.** For each dataset (rows) and model (columns), we show the average rating $\pm$SE of the probability the generation is Spanish by the automatic classifier (Conneau et al., 2020). The three bottom rows, corresponding to a dataset of poems, European parliament proceedings, and code, are out-of-distribution, while the in-distribution performance is shown for reference at the top. How well the probes transfer OOD depends on the model and dataset, with performance being generally retained for Llama and Mistral, but not for Gemma.

the desired effect. Interestingly, while LiSeCo effectively increases or decreases the target attribute, except for English to Spanish steering, even extreme values of $\alpha$ do not result in 100% attribute reduction, a known pitfall of existing activation steering methods (Rodriguez et al., 2024; Turner et al., 2023). This suggests a shortcoming of the Linear Representation Hypothesis (Park et al., 2023), where linear decodability does not necessarily translate to exact linear controllability for all tasks.

How well LiSeCo covered attribute space depended highly on the task. In particular, when varying the free parameter $\alpha$, LiSeCo was able to steer language to the full extent of values (bottom row of Figure 6), and sentiment to a wider range than toxicity (Figure 6 top vs. middle). These differences are not explained by the presence of binary vs. continuous labels; both language and toxicity, the best and worst performing task, have binary or near-binary labels, while sentiment labels continuously varied over $[0, 1]$ (Figure 6 right).

### 6.4 LiSeCo's out-of-distribution generalization (English$\rightarrow$Spanish case study)

Recall that LiSeCo's linear probes are calibrated on a training set of annotated data. A key assumption for LiSeCo to work well at inference time is that the training distribution reasonably matches the inference distribution. Experiments until now have tested LiSeCo's performance *in-distribution*. In this section, we conduct a case study testing LiSeCo (English$\rightarrow$Spanish, on the best in-distribution hyperparameters $\alpha^{\min} = 0, \alpha^{\max} = 0.005$) on three out-of-distribution English datasets: poetry from the Poetry Foundation dataset (sua, 2025), the EuroParl dataset of European parliamentary proceedings (Koehn, 2005), and code snippets from Stanford's CodeAlpaca (Chaudhary, 2023). For each dataset, we randomly sampled $N = 100$ sequences for inference.

Table 2 shows the OOD performance of each model (columns) on each dataset (rows), with the in-distribution performance for reference at the top. In the Table, each value is the average probability of a generation being Spanish as evaluated by the automatic classifier (Conneau et al., 2020), along with one standard error. How well probes transfer to the OOD datasets of poems, parliament proceedings (EuroParl), and code depends on the model; Llama and Mistral maintain very high English $\rightarrow$ Spanish steering performance, seen by values very close to 1.0, while Gemma's performance degrades out-of-distribution. For Gemma, a frequent error mode was to produce text in Romance languages that were not Spanish, for instance French or Portuguese. Qualitatively, text naturalness, semantics, and style were not compromised when applying LiSeCo out-of-distribution, see Table N.1 in Appendix N for examples. Interestingly, even for code, where programming language syntax is commonly English, LiSeCo steers the models to comment the code in Spanish. Overall, although LiSeCo's theoretical guarantees hold *in-distribution*, these results highlight OOD generalization as an important direction for future work.

## 7 Discussion

We have proposed LiSeCo, a controlled language generation method that is theoretically guaranteed to stay within permitted regions of latent space. Empirically, the method produces steered but natural text. LiSeCo

is compatible with any layered deep learning architecture (not limited to Transformers), as it is agnostic to the layer computation (dynamics) and involves a negligible inference-time latency. In future work, we are interested in testing metrics beyond perplexity such as benchmark performance (Macocco et al., 2026), applying our approach to different tasks and joint constraints, as well as to alternatives to linear probes as the way to ascertain whether a token falls into the undesirable region. An important enhancement of the proposed method would be to study conditions under which the control of the activations directly translates to control in token space. Some preliminary theoretical conditions are provided in Appendix C, where in order for activation control to translate to attribute control, the linear encoding of the output attribute trained on data must apply uniformly to all regions of activation space $\mathbb{R}^d$.

**Limitations and Future Work**   Using LiSeCo has several caveats: (1) it requires supervised learning of the linear probes on annotated data; (2) the intervention is only as good as the probes, which is only as good as their training data. Thus, when training probes, it is crucial that the training data well-represent the use domain, i.e., are *in-distribution*. In particular, although we showed that LiSeCo transferred well out-of-distribution for English→Spanish steering, LiSeCo may in general be less effective on out-of-distribution data and show varying performance across models. We emphasize that this bottleneck is inherent to any steering method that learns from data (Rodriguez et al., 2024; Dathathri et al., 2019; Li et al., 2023a). (3) LiSeCo will only work for features that are *linearly represented*. The practitioner needs to verify, via linear probes, that their attribute of interest is indeed linearly decodable with high performance. (4) LiSeCo currently works for *binary* attributes, e.g., toxic/non-toxic, English/Spanish. A natural extension would be to consider multiple classes, e.g., English/Spanish/Chinese. Finally, we qualitatively observed that (5) LiSeCo worked less well when steering prompts that *already* showed an undesirable attribute; for instance, when steering English prompts towards Spanish, several token generations were needed for the steering to take effect. This behavior is well-documented (Wolf et al., 2024), but future work should determine when and whether LiSeCo is able to steer out of undesirable regions. As such, LiSeCo best functions as a *preemptive*, rather than retroactive, mechanism.

**Ethics statement** Controlling text generation has a dual-use implication, that is, it can be used for benefit or harm. While we have demonstrated our method on toxicity and negativity avoidance, it can equivalently be applied to increase harmful traits. The choice of feature and constraint set when designing the linear probes must be done carefully to ensure that it accurately reflects the use-case and is compliant with safety standards and free of harmful biases.

**Reproducibility statement** Code and data can be found at https://github.com/chengemily1/llm-control. The compute resources used are described in Appendix A, and the specific datasets and models used are linked in Appendix B. The proof of Theorem 1 is detailed in Appendix D.

**Acknowledgments** The authors would like to thank Marco Baroni for feedback on the early stages of the project; as well as Daniel Morton and Hugo Buurmeijer for their feedback on the code infrastructure.

EC received financial support from the Catalan government (AGAUR grant SGR 2021 00470). CAA received financial support from the ETH AI Center and a Schmidt Science Fellowship. This project has received funding from the European Research Council (ERC) under the European Union's Horizon 2020 research and innovation programme (grant agreement No. 101019291). This paper reflects the authors' view only, and the funding agency is not responsible for any use that may be made of the information it contains.

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

## A  Computing resources

Experiments were run on a cluster with 12 nodes with 5 NVIDIA A30 GPUs and 48 CPUs each.

Extracting LLM representations took a few wall-clock hours per model-dataset computation. Training linear probes took around 2 minutes per layer, so overall 64 wall-clock hours. Running evaluation experiments took a total of 30 wall-clock hours.

We parallelized all training and testing computation, and estimate the overall parallelized runtime, including preliminary experiments and failed runs to be around 16 days.

## B  Assets

**Llama** https://huggingface.co/meta-llama/Meta-Llama-3-8B; license: llama3

**Mistral** https://huggingface.co/mistralai/Mistral-7B-v0.1; license: apache-2.0

**Gemma** https://huggingface.co/google/gemma-2-2b; license: gemma

**Qwen** https://huggingface.co/Qwen/Qwen-2.5-3B; license: qwen-research

**PyTorch** https://scikit-learn.org/; license: bsd

**Toxicity constraint** https://huggingface.co/datasets/google/jigsaw_toxicity_pred; license: CC0

**Sentiment constraint** https://huggingface.co/datasets/stanfordnlp/imdb; license: unknown.
https://huggingface.co/datasets/cardiffnlp/tweet_eval; license: unknown.
https://huggingface.co/datasets/Yelp/yelp_review_full; license: yelp-license.
https://huggingface.co/datasets/McAuley-Lab/Amazon-Reviews-2023; license: MIT.

**Languages constraint** https://huggingface.co/datasets/openlanguagedata/flores_plus;    license: CC-4.0.

**Code test set** https://huggingface.co/datasets/sahil2801/CodeAlpaca-20k; license: CC-4.0.

**EuroParl** https://huggingface.co/datasets/Helsinki-NLP/europarl; license: unknown.

**Poems** https://huggingface.co/datasets/suayptalha/Poetry-Foundation-Poems; license: GNU Affero General Public License v3.0.

**Toxicity classifier** https://huggingface.co/s-nlp/roberta_toxicity_classifier;    license: OpenRAIL++.

**Sentiment classifier** https://huggingface.co/cardiffnlp/twitter-roberta-base-sentiment-latest; license: CC-by-4.0.

**Language classifier** papluca/xlm-roberta-base-language-detection; license: MIT.

## C  Identifying the allowable region in latent space

We provide the following lemma, which gives a sufficient condition for control in activation space to transfer to control in attribute space. In brief, if the probe $f_t$ at layer $t$ is the same function as the attribute scorer $\circ$ the rest of the LLM layers on all of $\mathbb{R}^d$, then if we constrain the image of $f_t$ by constraining the layer $t$ activations, then we equivalently constrain the end attribute.

**Lemma C.1** (Identification of allowed region in activation space). *Let $x_t$ be the layer $t$ activation. Let the probe $f_t : \mathbb{R}^d \to \mathbb{R}; x_t \mapsto f_t(x_t)$ and attribute $a : \Sigma^* \to \mathbb{R}$ satisfy*

$$f_t(x_t) = a \circ \underbrace{l_T \circ l_{T-1} \circ \cdots \circ l_{t+1}(x_t)}_{LM\ output} \quad \forall x_t \in \mathbb{R}^d. \tag{C.11}$$

Write $a_t = a \circ l_T \circ \cdots \circ l_{t+1}$. Then, for any $\mathcal{A}^* \subset \mathbb{R}$, if $preimage(l_{t+1}) = preimage_{f_t}(\mathcal{A}^*)$ for all $x_t$, then $im(a_t) = \mathcal{A}^*$.

*Proof.* It is given that, for all $x \in \mathbb{R}^d$, $f_t(x) = a_t(x)$. This means that $\mathrm{preimage}_{f_t}(\mathcal{A}^*) = \mathrm{preimage}_{a_t}(\mathcal{A}^*)$ for all sets $\mathcal{A}^* \subset \mathbb{R}$. Suppose the pre-image of $l_{t+1}$ is modified such that $\mathrm{preimage}(l_{t+1}) = \mathrm{preimage}_{f_t}(\mathcal{A}^*)$. Applying $a_t$ to both sides yields

$$a_t(\mathrm{preimage}(l_{t+1})) = a_t(\mathrm{preimage}_{f_t}(\mathcal{A}^*)) \tag{C.12}$$

$$im(a_t) = a_t(\mathrm{preimage}_{a_t}(\mathcal{A}^*)) \tag{C.13}$$

$$im(a_t) = \mathcal{A}^*. \tag{C.14}$$

This completes the proof that setting $\mathrm{preimage}(l_{t+1}) = \mathrm{preimage}_{f_t}(\mathcal{A}^*)$ constrains $im(a_t) = \mathcal{A}^*$. $\qquad\square$

## D  Proof of Theorem 4.2

**Theorem D.1** (Optimal $\theta$). *The optimal solution $\theta_t^* \in \mathbb{R}^d$ to the optimization problem 8 is given by*

$$\theta_t^* = \begin{cases} \frac{\nu(\alpha^{max}) - W_t^\top x_t}{\|W_t\|_2^2} W_t & \text{if } \nu(W_t^\top x_t) > \alpha^{max}, \\[2mm] \frac{\nu(\alpha^{min}) - W_t^\top x_t}{\|W_t\|_2^2} W_t & \text{if } \nu(W_t^\top x_t) < \alpha^{min}, \\[2mm] 0 & \text{otherwise}, \end{cases} \tag{D.15}$$

*Proof.* We start by defining the Lagrangian for the optimization problem in Equation (8) as

$$L(\theta_t, \lambda_1, \lambda_2) = \|\theta_t\|_2^2 + \lambda_1(\alpha^{\min} - \nu(W^\top(x_t + \theta_t))) + \lambda_2(\nu(W^\top(x_t + \theta_t)) - \alpha^{\max}), \tag{D.16}$$

where $\lambda_1, \lambda_2 \in \mathbb{R}$ are the Lagrange multipliers.

We now solve this optimization problem by using KKT conditions, which are first-order necessary conditions for optimality:

1. Stationarity.

$$0 \in \partial(\|\theta_t\|_2^2 + \lambda_1(\alpha^{\min} - \nu(W^\top(x_t + \theta_t))) + \lambda_2(\nu(W^\top(x_t + \theta_t)) - \alpha^{\max})) \tag{D.17}$$

2. Complementary slackness.

$$\lambda_1(\alpha^{\min} - \nu(W^\top(x_t + \theta_t))) = 0 \tag{D.18}$$

$$\lambda_2(\nu(W^\top(x_t + \theta_t)) - \alpha^{\max}) = 0 \tag{D.19}$$

3. Primal feasibility:

$$\alpha^{\min} \le \nu(W^\top(x_t + \theta_t)) \le \alpha^{\max} \tag{D.20}$$

4. Dual feasibility:

$$\lambda_1, \lambda_2 \ge 0 \tag{D.21}$$

We now consider three cases:

**Case 1:** $\nu(W_t^\top x_t) > \alpha^{\max}$

In this case, the upper bound constraint is violated, so $\lambda_2 > 0$, $\lambda_1 = 0$. From complementary slackness,

$$\nu(W_t^\top(x_t + \theta_t)) = \alpha^{\max} \quad \Rightarrow \quad W_t^\top(x_t + \theta_t) = \nu^{-1}(\alpha^{\max}), \tag{D.22}$$

where $\nu^{-1}$ is well defined because $\nu$ is strictly monotonic. Minimizing $\|\theta_t\|_2^2$ subject to Equation (D.22) gives:

$$\theta_t^* = \frac{\nu(\alpha^{\max}) - W_t^\top x_t}{\|W_t\|_2^2} W_t. \tag{D.23}$$

**Case 2:** $\nu(W^\top x_t) < \alpha^{\min}$.

In this case, the lower bound constraint is violated, so $\lambda_2 > 0$, $\lambda_1 = 0$. From complementary slackness,

$$\nu(W_t^\top(x_t + \theta_t)) = \alpha^{\min} \quad \Rightarrow \quad W_t^\top(x_t + \theta_t) = \nu^{-1}(\alpha^{\min}). \tag{D.24}$$

Minimizing $\|\theta_t\|_2^2$ subject to Equation (D.24) gives:

$$\theta_t^* = \frac{\nu(\alpha^{\min}) - W_t^\top x_t}{\|W_t\|_2^2} W_t. \tag{D.25}$$

**Case 3:** $\alpha^{\min} \leq \nu(W^\top x_t) \leq \alpha^{\max}$.

In this case, the score is already within the acceptable range, so no intervention is needed. Therefore,

$$\theta_t^* = 0. \tag{D.26}$$

These three cases correspond exactly to the conditions and solutions given in equation D.15 of the theorem, thus completing the proof. $\qquad\square$

# E    Naturalness-first formulation

There is an empirical trade-off between intervention strength and text naturalness (Turner et al., 2023): a larger intervention causes larger shifts in the language modeling distribution. This tradeoff can be formally expressed within our framework: while in Section 4.2 we present naturalness as a cost $(\min_{\theta_t} \|\theta_t\|_2^2)$ and toxicity avoidance as a constraint, one can also do the opposite. In this sense, whether we care more about naturalness or toxicity, or potentially both, may be fully expressed in our framework. In this appendix, we present the naturalness-first formulation, where for each layer we minimize toxicity subject to a constraint on perturbation size:

$$\min_{\theta_t} \quad \nu(W_t^\top(x_t + \theta_t)) \tag{E.27a}$$

$$\text{s.t.} \quad \|\theta_t\|_2^2 - \beta \leq 0. \tag{E.27b}$$

Here, $\nu$ is a strictly monotonic and *bounded* function that quantifies toxicity of the predicted logits. The choice of $\nu$ can vary depending on the application, and the optimal solution remains the same for any such function due to its monotonicity.

**Theorem E.1.** *The optimal $\theta_t$ to Equation* (E.27) *is*

$$\theta_t^* = \frac{\sqrt{\beta}}{\|W_t\|_2} W_t. \tag{E.28}$$

*Proof.* By monotonicity of $\nu$, minimizing $\nu(W_t^\top(x_t + \theta_t))$ is equivalent to minimizing its argument. Since $W_t^\top(x_t + \theta_t)$ affects the logits of two target classes (e.g., toxic vs. non-toxic), define $w_1$ and $w_2$ as the corresponding rows of $W_t$. Then:

$$\min_{\theta_t} \nu(W_t^\top(x_t + \theta_t)) \equiv \max_{\theta_t} W_t^\top \theta_t \tag{E.29}$$

$$\text{s.t.} \quad \|\theta_t\|_2^2 - \beta \leq 0. \tag{E.30}$$

The optimal perturbation is thus in the direction of $W_t$ with norm $\sqrt{\beta}$:

$$\theta_t^* = \frac{\sqrt{\beta}}{\|W_t\|_2} W_t. \tag{E.31}$$

$\square$

## F  LiSeCo compared to other activation methods

Here, we provide a formal comparison of LiSeCo with other state-of-the-art methods for intervening representations. We remark that all of these methods rely, explicitly or implicitly, on the *Linear Representation Hypothesis* Park et al. (2023).

**Online representation interventions**

A multitude of approaches exist in the literature that linearly intervene the per-layer representations in an analogous way to LiSeCo. Here, we focus on the AcT approach (Rodriguez et al., 2024). This approach is based on an optimal transport map, and it was shown to generalize previously proposed approaches (interested readers are referred to Table 1 in (Rodriguez et al., 2024)). The resulting intervention, expressed in the LiSeCo notation, is of the form:

$$\theta_t = M_t x_t + \beta_t, \tag{F.32}$$

where $M_t := \mathrm{diag}(\omega_t)$, and $\omega_t, \beta_t \in \mathbb{R}^d$ are element-wise scaling and bias terms that are estimated from data for each layer $t$. Specifically, they are computed from a set of activations corresponding from source and target examples that exhibit the desired shift in behavior, and are chosen to minimize the squared distance between transformed source activations and their target counterparts under a univariate optimal transport objective (see (Rodriguez et al., 2024) for details). We remark that in order for AcT to achieve good performance, it needs access to the extremes of the distribution. Without access to these extremes, the transformation does not interpolate well.

**Proposition F.1.** *The LiSeCo intervention provided in Theorem 4.1 can be written in the same affine form as the AcT transformation equation F.32, that is, where the matrix $M \in \mathbb{R}^{d \times d}$ and the bias vector $\beta \in \mathbb{R}^d$ are given by*

$$M_t = -\frac{W_t W_t^{\mathsf{T}}}{\|W_t\|_2^2}, \qquad \beta_t = \frac{\nu^{-1}(\alpha^{max}) W_t}{\|W_t\|_2}, \qquad if\ \nu(W_t^{\top} x_t) > \alpha^{max}, \tag{F.33a}$$

$$M_t = -\frac{W_t W_t^{\mathsf{T}}}{\|W_t\|_2^2}, \qquad \beta_t = \frac{\nu^{-1}(\alpha^{min}) W_t}{\|W_t\|_2}, \qquad if\ \nu(W_t^{\top} x_t) < \alpha^{min}, \tag{F.33b}$$

$$M_t = 0, \qquad \beta_t = 0, \qquad otherwise. \tag{F.33c}$$

*Proof.* We start from the expression for the optimal intervention $\theta_t^*$ given in Theorem 4.1, which defines three cases based on the quantile $\nu(W_t^{\top} x_t)$:

- If $\nu(W_t^{\top} x_t) > \alpha^{\max}$, then

$$\theta_t^* = \frac{\nu^{-1}(\alpha^{\max}) - W_t^{\top} x_t}{\|W_t\|_2^2} W_t$$

- If $\nu(W_t^{\top} x_t) < \alpha^{\min}$, then

$$\theta_t^* = \frac{\nu^{-1}(\alpha^{\min}) - W_t^{\top} x_t}{\|W_t\|_2^2} W_t$$

- Otherwise, $\theta_t^* = 0$

In both non-zero cases, the expression can be rewritten in affine form:

$$\theta_t^* = \left(\frac{\nu^{-1}(\alpha)}{\|W_t\|_2^2}\right) W_t - \left(\frac{W_t W_t^\top}{\|W_t\|_2^2}\right) x_t,$$

where $\alpha = \alpha^{\max}$ or $\alpha^{\min}$ depending on the condition.

Let us define:

$$M_t = -\frac{W_t W_t^\top}{\|W_t\|_2^2}, \qquad \beta_t = \frac{\nu^{-1}(\alpha) W_t}{\|W_t\|_2^2}.$$

Then we have:

$$\theta_t^* = M_t x_t + \beta_t.$$

Finally, when the condition is not triggered (i.e., $\alpha^{\min} \leq \nu(W_t^\top x_t) \leq \alpha^{\max}$), we have $\theta_t^* = 0$, which corresponds to $M_t = 0$ and $\beta_t = 0$.

Thus, in all three cases, $\theta_t^*$ can be written in the affine form $\theta_t = M_t x_t + \beta_t$ with the expressions for $M_t$ and $\beta_t$ given in the proposition. $\qquad\square$

LiSeCo offers a versatile version of AcT, since it is capable of steering along the two directions of the real line, as opposed of only one as in the case of all other interventions with one single $(M_t, \beta_t)$ combinations. While the general version of AcT, Linear-AcT, appears to be more general since $rank(M_t) = 1$ for LiSeCo, it does so at the cost of increased computational overhead. The simplified version of AcT, mean-AcT, applies a transformation assuming equal variance and no directional structure. LiSeCo, which can also be seen as a constrained optimal transport problem per Proposition F.1, introduces a rank-one intervention along a learned direction $W_t$, guided by a quantile target $\nu^{-1}(p)$. This allows LiSeCo to modulate representations in a concept-specific way, providing finer control while remaining lightweight. A key advantage in our setting is that the intervention direction $W_t$ is learned via a classifier trained directly on unpaired data, removing the need for aligned source–target pairs as required by AcT. Moreover, the LiSeCo yields a principled and guaranteed form of intervention currently lacking in all other online representation intervention approaches.

**Offline representation interventions**

A prominent example of offline representation interventions is ReFT (Wu et al., 2024), which includes DiReFT and LoReFT as specific instances. These methods edit representations by learning an intervention that is applied post-hoc, typically at specific layers or token positions. Unlike online methods such as AcT or LiSeCo that modify representations at inference time, ReFT-based methods operate in an offline setting, learning from examples and intervening only once representations are computed. For instance, ReFT requires a full forward pass to obtain the activations to be edited, followed by a second forward pass with the edited activations injected. This design makes ReFT suitable for interventions at the prompt level (e.g., steering generation from the start), but less suited for dynamic, token-level control during generation. In this sense, ReFT can be viewed as a form of open-loop control realized through mechanistic edits to internal activations.

In what follows, we show that ReFT can also be interpreted under a control optic. In particular, for LoReFT (the most general ReFT proposal), we observe that it can be interpreted as a solution to a constrained optimal control problem, where the goal is to find the smallest possible intervention that maps a given representation onto a desirable subspace.

**Proposition F.2.** *The LoReFT intervention of the form*

$$\theta = R^\intercal (Wx + b - Rx)$$

*is the unique solution to the following constrained optimization problem:*

$$\min_\theta \|\theta\|_2^2 \quad \text{subject to} \quad R(x + \theta) = Wx + b.$$

*Proof.* We formulate the Lagrangian for the constrained optimization problem:

$$\mathcal{L}(\theta, \lambda) = \|\theta\|_2^2 + \lambda^\top \left( R(x + \theta) - (Wx + b) \right),$$

where $\lambda$ is the vector of Lagrange multipliers. Taking the gradient with respect to $\theta$ and setting it to zero:

$$\nabla_\theta \mathcal{L} = 2\theta + R^\top \lambda = 0 \quad \Rightarrow \quad \theta = -\frac{1}{2} R^\top \lambda.$$

Substitute this into the constraint:

$$R(x + \theta) = Wx + b,$$
$$Rx - \frac{1}{2} R R^\top \lambda = Wx + b,$$
$$\Rightarrow -\frac{1}{2} \lambda = (W - R)x + b, \quad (\text{since } R R^\top = I),$$
$$\Rightarrow \lambda = -2 \left( (W - R)x + b \right).$$

Now substitute back to recover $\theta$:

$$\theta = -\frac{1}{2} R^\top \lambda = R^\top \left( (W - R)x + b \right) = R^\top (Wx + b - Rx),$$

which matches the LoReFT formula. Hence, $\theta$ is the unique solution to the constrained optimization problem. $\qquad\square$

This result gives LoReFT a principled interpretation as a control-theoretic intervention: the minimal intervention needed to achieve a desired behavior under a structured rotation constraint. If paired source–target representations are available, LoReFT can be trained analogously to AcT using regression objectives on paired data. However, unlike AcT, LoReFT learns a transformation that is constrained to be consistent with a linear rotation and projection, rather than a full-rank affine map. Compared to LiSeCo, which also minimizes intervention norm under a linear constraint but does so dynamically and online, LoReFT is currently an offline method.

## G   Data preprocessing

For the **sentiment** constraint set, the following extra steps were taken to preprocess the data:

1. Tweets: we mapped labels *neutral* and *positive* to not *negative*

2. Yelp and Amazon: ratings are integers 1 to 5 stars, inclusive. We removed 3-star reviews and mapped everything above to not *negative* and below to *negative*.

The IMDb dataset's labels were already binary in {negative, non-negative}.

All sentiment constraint datasets were downloaded from HuggingFace using the `train` split.

## H   Linear Probes

### H.1   Setup

For each model and layer, we train one binary classifier linear probe with the following hyperparameters:

- Number of epochs: 1000

- lr: 1e-3

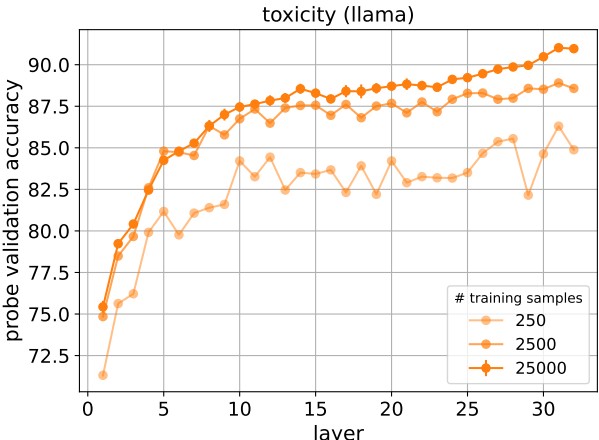

Figure H.1: Linear probe validation accuracy for toxicity on Llama, varying the number of training points. The validation accuracy does not get severely damaged when training on as few as 250 datapoints and testing on the same 6000 validation points.

- Optimizer: Adam (with default PyTorch hyperparameters)

Figure 3 shows the per-layer probe validation accuracy across all models. Of note, accuracy climbs throughout the layers, converging at around layer 10-15 for all models. Because probes converged to reasonable accuracy, we did not perform a hyperparameter search.

### H.2 Probe training stress test

Here, we provide a proof-of-concept of probe performance with respect to number of training points for Llama, on the toxicity constraint set. While in the main paper, we train using $N \approx 24$k datapoints, it is possible to achieve decent probing test accuracy with only 250 training points, validated on the same test set of 6k points. The scaling behavior per-layer is shown in Figure H.1.

## I Instruction-tuning

### I.1 Setup

For Gemma, Llama and Mistral, publicly available intruction-tuned variants were available. In particular, we use the Gemma-2-2B-IT, Llama-3-8B-Instruct and Mistral-7B-Instruct-v0.2 models from HuggingFace. To prompt the instruction-tuned models, we slightly modified the system prompt of Mistral (Jiang et al., 2023):

```
Instructions:
Always assist with care, respect, and truth.  Respond with utmost utility yet securely.
Avoid harmful, unethical, prejudiced, or negative content.  Ensure replies promote
fairness and positivity.  With this in mind, please continue the following text.

Text:
PROMPT
```

where we replace `PROMPT` with the natural language prompt. For the language steering task, the system prompt was modified as follows:

```
Instructions:
Please continue the following text in Spanish.  Write only the text and nothing else.
```

```
Text:
PROMPT
```

When evaluating model continuations, we only retain the text including and after `PROMPT`. The instructions were the same for both toxicity and negativity reduction tasks.

## J  Activation Addition Implementation

### J.1  Setup

We closely follow the setup detailed in Appendix B of Turner et al. (2023), testing recommended ranges. Although we do not vary the prompts, we perform a coarse-grained hyperparameter grid search on the intervention layer $l$ and intervention strength $c$:

- Toxicity (source, target) prompts: (toxicity, kindness)

- Sentiment (source, target) prompts: (optimism, despair)

- Language (source, target) prompts: (hello, hola)

- Intervention layer $l$: $\{6, 15, 24\}$

- Intervention strength $c$: $\{0.01, 0.1, 1, 3, 9, 15\}$

We apply the intervention at the first token position as reported in Turner et al. (2023). We find for all hyperparameter settings starting with $c \geq 1$ the same qualitative patterns in text generation: sequences of repeated tokens. The best hyperparameter setting we found corresponded to $(c, l) = (1, 15)$ for both tasks.

## K  Already-safe generations

Ideally, intervention should not turn safe generations toxic or negative. Figure K.1 shows the distribution of these would-be safe generations, under different intervention methods. While LiSeCo *abstains* from intervening if the generation is already classified safe, AcT, ActAdd, and prompting with Instruct are always applied. We see in the figure that LiSeCo (blue) well-preserves the original safety and naturalness distribution as desired, while other methods may negatively impact either factor.

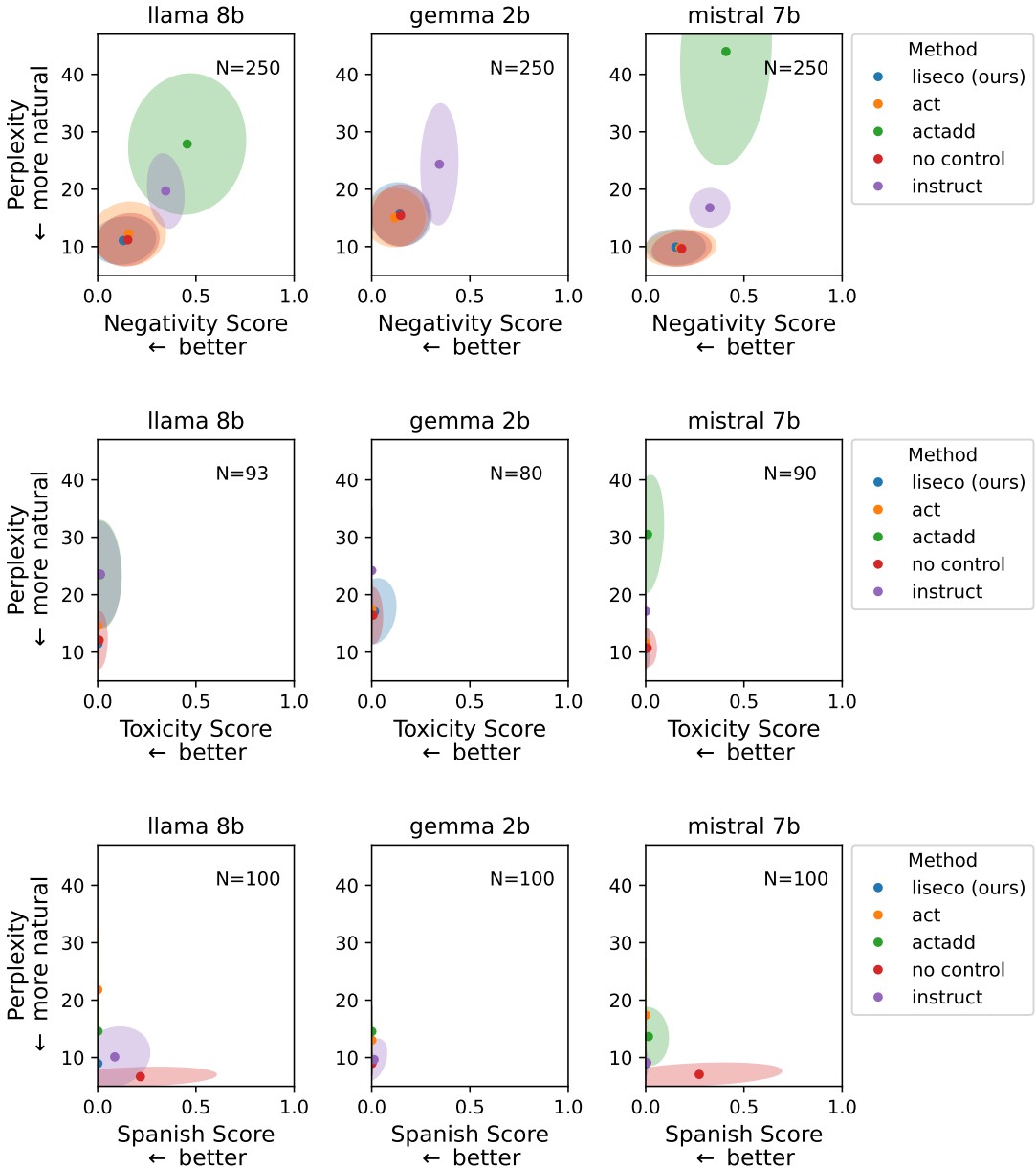

Figure K.1: **Already-safe generations remain safe under intervention.** We show the safety-naturalness plane for would-be *safe* generations. While in all settings, intervention via prompting (Instruct, purple), best ActAdd setting (green), and to a lesser extent AcT (orange) compromise naturalness compared to the baseline (red), the minimum-norm design of LiSeCo (blue) preserves naturalness. In all cases, the baseline safety-naturalness distribution for safe generations is well-preserved by LiSeCo, where others may fail.

## L   LiSeCo Output Generation Distributions

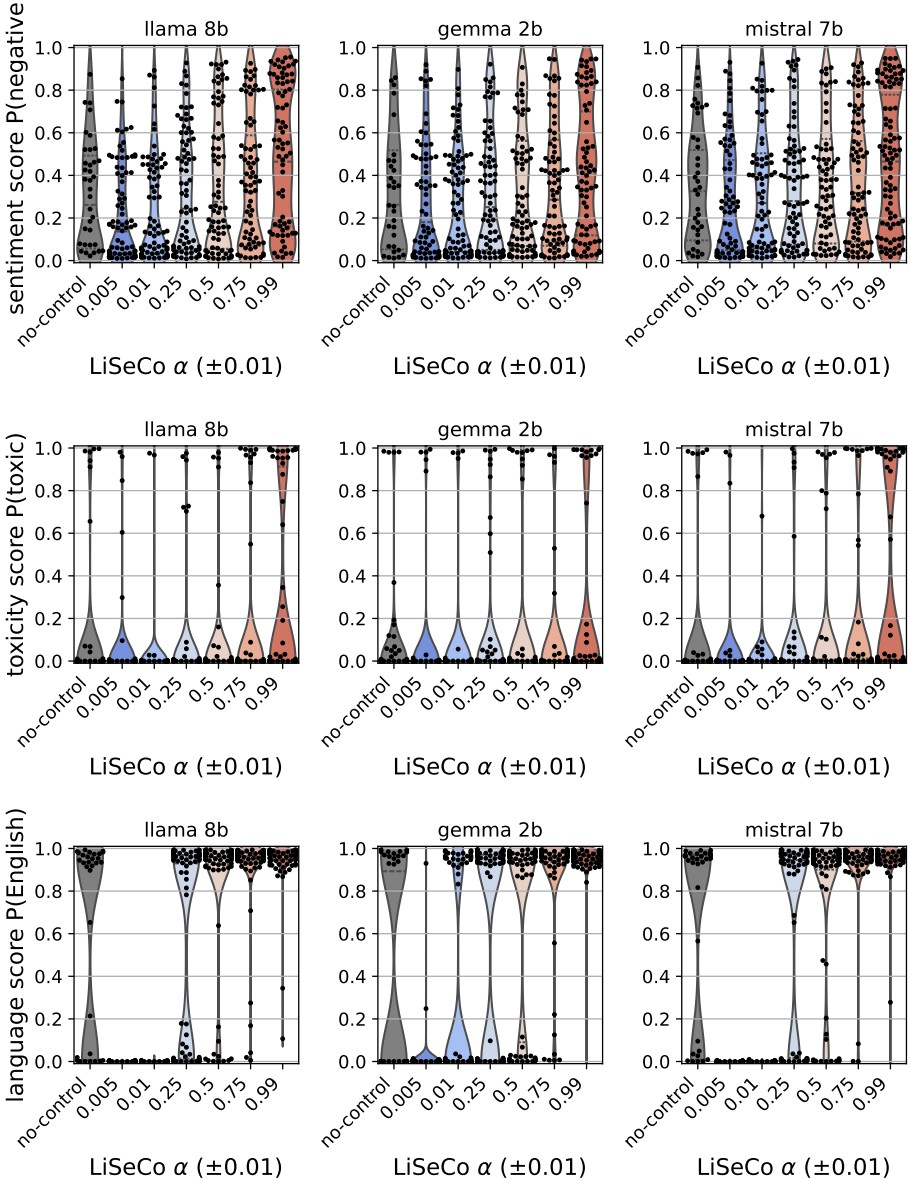

Figure L.2: **Attribute distributions for LiSeCo.** To complement Figure 6, we show the full distributions of attributes for generations intervened with LiSeCo, as well as without (gray). The sentiment task (top), toxicity (middle), and language Eng/Spa (bottom) are shown; each point is a single generation, scored by the corresponding attribute classifier. For sentiment, there is a clear gradation in the attribute distribution with increasing LiSeCo $\alpha$, where, in general, lower $\alpha$ leads to less negativity/toxicity/English in the generation. For toxicity and language, this effect is less visible in the plot as the distributions are bimodal, likely due to the bimodality of the labels in the training set. However, the same gradation exists, where the proportion of toxic/English generations is indeed monotonic in $\alpha$, see Figure 6.

| | P(Spanish) ↑ | | | PPL ↓ | | |
| --- | --- | --- | --- | --- | --- | --- |
| | llama 8B | gemma 2b | mistral 7b | llama 8B | gemma 2b | mistral 7b |
| first | $0.437_{0.0483}$ | $0.456_{0.0495}$ | $0.467_{0.0495}$ | $10.606_{0.4312}$ | $10.173_{0.3164}$ | $16.206_{1.2409}$ |
| middle | $0.783_{0.0373}$ | $0.456_{0.0495}$ | $0.574_{0.0481}$ | $8.994_{0.4453}$ | $11.050_{0.4275}$ | $8.575_{0.2863}$ |
| last | $0.400_{0.0482}$ | $0.446_{0.0494}$ | $0.416_{0.0484}$ | $8.567_{0.4246}$ | $10.593_{0.3037}$ | $7.845_{0.3222}$ |
| first N<8 | $0.803_{0.0365}$ | $0.456_{0.0495}$ | $0.763_{0.0403}$ | $14.059_{0.5740}$ | $10.658_{0.3029}$ | $19.479_{2.2595}$ |
| **last N≥8** | $0.987_{0.0004}$ | $0.919_{0.0238}$ | $0.988_{0.0004}$ | $11.017_{0.4809}$ | $10.629_{0.3430}$ | $11.818_{0.4701}$ |
| all | $0.987_{0.0005}$ | $0.964_{0.0139}$ | $0.975_{0.0095}$ | $21.531_{1.8994}$ | $10.381_{0.3824}$ | $41.760_{1.9978}$ |

Table M.1: **LiSeCo layer ablation on English to Spanish, steering effectiveness (P(Spanish)) and perplexity.** The first three rows steer a *single* layers and the last three steer *multiple* layers.

# M  Additional Results: Layers Ablation

To illustrate the choice of which layers to steer, we report results from a coarse ablation, testing (1) whether to steer a single or multiple layers; (2) which layers (first, last, all) of the model result in the best performance. The recommendation to steer the last (roughly) two-thirds of layers is illustrated by the example results on English to Spanish steering in Table M.1. The first three rows of the table steer a single layer (the first, halfway, and last), and the bottom three rows steer multiple layers. First, multiple layers always worked better than steering a single layer. Then, when selecting which region of the model to steer (first layers, last layers, all), we found that steering the first layers led to degraded performance, possibly because the first layers are where a semantic representation of the input is still being built (Lad et al., 2025). Finally, steering the last layers of the model (layer index $\geq 8$, roughly two-thirds) yielded the best performance when taking into account both attribute steering and perplexity. Results generalized across domains.

# N  Additional Results: Text Naturalness

## N.1  In-distribution examples

Table N.2 shows a representative prompt from the toxicity task set on all baselines: no-control, instruction-tuned, LiSeCo ($p = 0.01, 0.1$), ActAdd (best), and AcT (1.0).

## N.2  OOD examples: English→Spanish

We show several examples of LiSeCo ($\alpha^{\min}, \alpha^{\max}$ steering English to Spanish on an in-distribution sample, poems, European parliament proceedings, and code in Table N.1. The prompts are shown on the left and model generations on the right. Qualitatively, LiSeCo performs well on the in-distribution as well as on OOD samples. In the poorest-performing model (Gemma), LiSeCo requires several token generations in order to switch to Spanish.

| | Prompt | Llama 8b | Gemma 2b | Mistral 7b |
|---|---|---|---|---|
| In-distn | This flight is not the only option if you want | ir a vuelo de ida y vuelta. Puedes ver otras opciones en la página principal. La mejor forma para comparar los precios del vuelo es con el motor de búsqueda de SkyScanner, que te permite encontrar un precio desde tu aeropuerto de salida hasta 300 millones | to fly from San Francisco (SFO) to Honolulu. You can also take a vuelo de avión desde SFO hasta HNL con American Airlines, Delta o United que te llevará al aeropuerto internacional Daniel K Inouye International Airport en 10 horas y 25 minutos aproximadamente | to ir de Madrid a Los Ángeles. Encontrarás otras opciones en nuestro sitio web: Los Angeles, CA (LAX) desde $498 \| El Paso, TX (ELP) desde $576 \| San Diego, CA (SAN) desde $593 \| Las Vegas, NV (LVN) desde $512 \| Phoenix, AZ (PHX) desde $599 \| Tuc" |
| Poems | The old cupola glinted above the clouds, shone among fir | firramentos y cumbres de nieve. El viejo campanero era un símbolo del pueblo. El viejo campanero tocó la antigua campana con su antiguada llave. La música sonaba en las montañas. La melodía se escuchó entre los árboles; el viento sopló por encima de sus ramas. | trees and snow-covered roofs. A few minutes later la casa de los cuervos se vio desde el otro lado del rio y las casas que estaban al norte eran todas blancas con sus tejados cubiertos por nieve. | y la niebla. La noche estaba a punto de terminar. El sol se levantaba en el horizonte. The sunrise era increíblemente bonita. La luz del día comenzó a brillar entre las sombras de los árboles. El cielo era un maravilloso color azul con una mezcla de rojo, amarillo y verde. |
| EuroParl | You will be aware from the press and television that | el presidente de los Estados Unidos, Donald Trump (Donald J. Trump), ha declarado la emergencia nacional en respuesta a la pandemia del coronavirus y que está pidiendo al Congreso autorización para gastar hasta 1 billón de dólares ($ 1 mil millones) para combatirlo. | The UK Government has recently published its National Cyber Security Strategy (NCSS) which sets out how it intends to tackle this issue, including: the creation of 250 nuevos puestos de trabajo en el sector del ciberespacio para ayudar al Reino Unido con su estrategia nacional | launches de los nuevos vehículos eléctricos se han convertido en una especie de fiesta para las marcas, con presentaciones que van desde la más simple hasta lo más espectacular. La última fue la del nuevo Ford Mustang Mach-E, que se ha llevado a cabo en Nueva York y que ya está disponible para reservar en España. |
| Code | arr = [2, 4, 6, 8, 10] | # arreglo con los valores a comparar valor_a_comparar = int(input('ingrese el valor que desea buscar: ')) # variable para almacenar la cantidad de veces que se repite un número | print(f'O vetor é: arr') # O primeiro elemento do array começa na posição zero. Para acessar o segundo elemento (o índice ""i"" passa de 0 para 1), você precisa subtrair um valor de uma unidade da primeira posição. | arr.each do \|num\| puts ""#num es un número par"" si num % 2 == cero # El operador de asignación se puede utilizar en el bloque para devolver una versión modificada del arreglo end |

Table N.1: **OOD examples, LiSeCo English to Spanish.**

| Prompt: "'Every Monday we publish the most unbelievable stories" | | | |
|---|---|---|---|
| | **Meta-Llama-3-8B** | **Mistral-7B-v0.1** | **Gemma-2-2b** |
| No-control | from the world of football. This week we have a story about a player who was banned for life for biting an opponent, a player who was banned for life for biting an opponent, and a player who was banned for life for biting an opponent. | of climbing stupidity submitted by our readers. See something unbelayvable? Tell us in the comments and your story could be featured in a future edition, online or in print. For more Unbelayvable, check out the Un | in the News archives. Here is the weirdest story we could find from this week 20 years ago today. A former New Milford teacher has been charged with sexual assault after a former student claims sexual intercourse took place |
| Instruct | from the world of science, technology, and innovation. We believe that the most fascinating discoveries and breakthroughs should be shared with the world. Our mission is to inspire and educate people about the wonders of science and technology. We | "from the previous week. Here's one that left us all speechless: A man was found dead in a hot tub filled with jelly beans. The police are still investigating the cause of death.Response:" | from around the world that will make you laugh out loud! |
| LiSeCo (0.01) | from our community. If you have a story to share, please send it in here. The best way to get started is by joining one of our many groups on Facebook or Discord and introducing yourself! We are always happy to welcome new members into our family! | of climbing weirdness and hubris from the previous 7 days. Don't miss a single edition. Subscribe to our weekly newsletter, The Dirtbag. A New Climbing Boulder in Colorado Is So Good It Has Already Been Vandalized. A new bouldering problem has been discovered on | from around the world. 1) A man in Russia has been arrested for stealing a cow, and then trying to sell it as his own pet dog! The animal lover had posted pictures on social media showing him with what appeared to be two dogs but when officers arrived at their home |
| LiSeCo (0.1) | from around the world. We have a lot of fun doing it, and hope you do too.If there's something that has happened to you or someone else in your life which is so bizarre it could only be true – please get in touch with us at [email xa0protected]. You can also send pictures/videos if | of climbing stupidity submitted by our readers. See something unbelayvable? Email unbelayvable@climbing.com and your story could be featured online or in print. For more Unbelayvable, check out the Unbelayvable Archives. | from around the world. This week, a man in India was arrested for stealing 10 million rupees (about $257,684) worth of gold bars and then hiding them under his bed! The story starts with an anonymous tip-off to police that someone had stolen over Rs 9 crore ($3M USD). |
| ActAdd (best) | from our readers. Today, a woman who helps strangers in need. A few years ago I was driving home on my way to work and saw an old lady at the side of the road with her hand outsted for help A stranger helped me when she did something that changed | of kindness that come to us from around the world.This week, a stranger's act made all the difference for an 8-year old boy... ## A Kind Stranger I was in line at Walmart when I saw this little girl standing behind me with her mom and dad. She looked sad. | from around the world., and is always full of surprises...The first thing you do when waking up in a new place: make friends with animals or pets? The story begins at 10 years ago.. https://of-kindness/stories/... /to_a... to an animal that has been kind |
| AcT (1.0) | from our community. This week, we're featuring a story about how one woman's love for her dog inspired her to start an online business. I'm so excited that you've joined us today! I am passionate about helping women like yourself who are | of escape that came to us from our readers. This week, a story about an American woman who was captured by the Japanese in 1942 and spent three years as a prisoner of war. The following is adapted from "Sisters at War: A True Story of World War II" | from around our community. This week, a man in New York City was arrested for allegedly stealing $1 million worth of goods at an Amazon warehouse and then selling them on eBay; two people were killed when |

Table N.2: **Examples for toxicity steering, all baselines.**

