# OpenReview forum: "LiSeCo: Linear Semantic Control for Language Generation"
_TMLR — Accepted by TMLR_

### Review · Reviewer_cWJc · 2025-11-13

**Summary Of Contributions:**

The manuscript proposes a unified taxonomy for linearly controlled language models, systems in which linear interventions, controllers, or structured transformations are applied to guide the behavior of large language models. The authors aim to synthesize the diverse literature on linear steering, feature editing, control matrices, and linear disentanglement into a coherent conceptual framework. The paper also includes illustrative experiments that demonstrate how specific linear controls impact generation quality, stability, and interpretability.

The work’s main strengths are the clarity with which the taxonomy is laid out, the comprehensive literature sweep, and the attempt to formalize a space of techniques that is currently fragmented. The presentation is well-organized, and the empirical sections give concrete intuition for how different control families behave.

The main weaknesses stem from two gaps. First, the manuscript does not articulate, with sufficient specificity, how this taxonomy improves upon or differs from prior surveys in controllable generation or linear interventions in mechanistic interpretability. Second, several claims about generality, practical utility, and theoretical cohesion would benefit from more thorough justification, particularly where the taxonomy implies normative structure in a domain that is still empirical and heterogeneous.

Overall, the contributions are promising and potentially useful but require stronger contextualization and deeper analysis to firmly establish novelty and significance.

**Audience:**

Yes

**Audience Explanation:**

Even with the limitations noted above, the topic is timely. Many researchers in controllable generation, interpretability, and model steering are actively exploring linear methods, and a well-structured taxonomy could be valuable. The manuscript’s ambition to organize emerging practices and provide conceptual scaffolding aligns with the interests of the TMLR community. With stronger evidence and clearer differentiation from prior surveys, the work could become a useful reference point.

**Broader Impact Concerns:**

There are no major ethical concerns. Techniques for model control can have dual-use implications, but the manuscript does not introduce new manipulative capabilities. A brief acknowledgement that control mechanisms can influence safety, bias, or potential misuse would strengthen the broader impact section, although it is not strictly required.

**Claims And Evidence:**

No

**Claims Explanation:**

The theoretical framing is sound, but several claims either lack rigorous justification or rely on selective evidence. For example, the manuscript asserts that linear controllers form a “coherent” and “principled” family of interventions, yet many techniques cited in the taxonomy differ substantially in assumptions, objectives, and empirical behavior.

The experimental demonstrations are helpful but too limited in scope to support broad generalization. In addition, methodological details in the empirical section, dataset choices, evaluation metrics, and statistical robustness are only lightly sketched, which makes it difficult to assess the strength of the conclusions. Strengthening the empirical methods and clarifying the scope of the claims would resolve these concerns.

**Requested Changes:**

1. Clarify novelty and differentiation from existing surveys.
2. Strengthen empirical evaluation. The current experiments are illustrative but not rigorous enough to substantiate the more general claims.
3. Avoid overgeneralized claims about cohesion of “linear control.” The taxonomy is useful, but several claims about unified structure should be tempered or supported by stronger evidence. Consider framing these as hypotheses or observed patterns rather than definitive properties.
4. Improve methodological detail. Clarify datasets, model configurations, sampling settings, evaluation metrics, and experiment reproducibility. This is especially important given the conceptual nature of the paper.

---

> ### Author Response · Authors · 2026-03-20
>
> We thank the reviewer for the thorough comments and suggestions.
>
> Regarding the points raised, we agree that existing activation-based methods that employ linear interventions differ substantially in their assumptions, objectives, and empirical behavior. Our use of the terms “coherent” and “principled” was not intended to suggest homogeneity along these dimensions. Rather, by coherent, we mean that these methods share a common structural form: they intervene via additive (linear) modifications to latent activations during generation. This shared structure permits a unified analysis in terms of state-space trajectories and admissible interventions, even when the optimization objectives or learning procedures differ. By principled, we refer specifically to the fact that linear interventions admit formal characterization and analysis, as opposed to heuristic or purely prompt-based approaches. Importantly, the absence of shared guarantees across prior methods is precisely the gap our work addresses: while previous approaches employ linear interventions implicitly or empirically, they do not formulate them as solutions to a control problem with explicit constraints and guarantees. To avoid ambiguity, we have revised the manuscript to clarify that our claim of coherence concerns the intervention class rather than the learning objective or empirical outcomes, and to more explicitly distinguish our control-theoretic formulation from prior linear steering methods.
>
> Below, we detail how we have specifically addressed the requested changes. We also provide an updated version of the manuscript, with blue highlights on the changes:

---

> > ### Author Response · Authors · 2026-03-20
> >
> > > Clarify novelty and differentiation from existing surveys.
> >
> > We have enhanced the contributions section of our method and included it in the introduction. Here, we highlight the differences of the proposed method (LiSeCo) as compared with other methods presented in the literature. In particular, we have made explicit the fact that our key novelty is conceptual rather than incremental: LiSeCo is the first work to formulate activation steering in language models as an optimal control problem. Existing steering approaches surveyed in the literature are primarily empirical, heuristic, or iterative, and do not provide formal guarantees. By contrast, our formulation admits a closed-form solution that provably enforces activation-level constraints, enabling exact, efficient control rather than approximate nudging. This optimal control framing yields three concrete advantages over prior work: (i) formal guarantees on activation compliance, as the intervention is derived to exactly satisfy hard constraints in representation space rather than encouraging them heuristically; (ii) minimal interventions, since the optimal solution has the smallest possible magnitude required to restore feasibility and therefore activates only when constraints would otherwise be violated, which avoids persistent or excessive steering and helps preserve the naturalness of the generated text; and (iii) low-latency inference, because the intervention is computed in closed form, eliminating the need for backpropagation, iterative optimization, or sampling-time auxiliary models used in prior methods. While existing surveys *document* empirical steering techniques, they do not address guaranteed, optimal, or analytically solvable control of activations. Instead, we provide a new method. LiSeCo therefore introduces a fundamentally different paradigm by treating steering as a constrained optimal control problem, which we now clearly emphasize in the revised manuscript.

---

> > > ### Author Response · Authors · 2026-03-20
> > >
> > > > Strengthen empirical evaluation. The current experiments are illustrative but not rigorous enough to substantiate the more general claims.
> > >
> > > To broaden the experimental section, we have added steering English -> Spanish (see updated Figures, where an additional row is added for language steering). LiSeCo works very well on this use case. We also added more experiments that stress-test LiSeCo on OOD generalization. LiSeCo is able to steer English->Spanish even on OOD data. Still, we emphasize that LiSeCo works best when applied in-distribution. We are happy to consider any specific experiments you suggest.

---

> > > > ### Author Response · Authors · 2026-03-20
> > > >
> > > > > Avoid overgeneralized claims about cohesion of “linear control.” The taxonomy is useful, but several claims about unified structure should be tempered or supported by stronger evidence. Consider framing these as hypotheses or observed patterns rather than definitive properties.
> > > >
> > > > Thank you for bringing this to our attention! We agree that claims about the cohesion of “linear control” should be stated carefully, and we have revised the manuscript to temper and clarify these statements. In particular, the term linearity is used in two distinct senses in the paper, which we now explicitly distinguish. First, by linear (activation-level) control we refer to the structural form of the intervention, namely additive modifications of the latent state of the form x↦x+θ. In this sense, we use “linear control” as a modeling abstraction: many activation-based steering methods can be viewed as instances of additive state perturbations, regardless of how θ is computed. This common structure motivates a shared analytical lens, but does not imply that these methods share objectives, guarantees, or empirical behavior. We have revised the text to emphasize this structural interpretation and to avoid overgeneralized claims of methodological unity. Second, linearity also arises in our representational assumptions, namely the use of linear classifiers to define admissible regions in embedding space. This relies on the linear separability hypothesis—that relevant semantic attributes can be approximately separated by linear probes in the latent space of the model. We now consistently frame this as a hypothesis rather than a definitive property, and explicitly cite prior work that adopts similar assumptions. Importantly, we empirically validate the relevance of this hypothesis in our setting: linear probes achieve high accuracy on the attributes considered (Figure 3), supporting the use of linear constraints for control in our experiments. Finally, in the taxonomy presented in Appendix F, we explicitly note that the unifying perspective we adopt is conditional on these structural and representational assumptions. Our intent is not to claim a definitive unified theory of activation steering, but rather to highlight observed patterns that motivate a control-theoretic formulation, which our method instantiates with explicit guarantees.

---

> ### Author Response · Authors · 2026-03-20
>
> > Improve methodological detail.
>
> The methods are described in detail. The exact Huggingface links and licenses of all models and datasets are given in Appendix B. Section 5.1.1 and Appendix G describe our dataset preprocessing steps. Section 5.1.2 and Appendix H describe the linear probe training procedure, with the exact hyperparameters. Appendix B provides the Huggingface links to the models we used to score generations. We also fully described the hyperparameters used for the baselines: Appendix J (ActAdd), in the main text for AcT (parameter = 1.0), Appendix I for Instruct. We are happy to clarify any specific questions you may have about the methods.

---

> ### Author Response · Authors · 2026-03-20
>
> >  A brief acknowledgement that control mechanisms can influence safety, bias, or potential misuse would strengthen the broader impact section, although it is not strictly required.
>
> Regarding the ethical considerations, we have modified our ethics statement on page 20 to reflect the points brought up by the reviewer.

---

### Review · Reviewer_JS8J · 2026-01-06

**Summary Of Contributions:**

This paper mainly focuses on the controlled language generation methods so that the large language models (LLMs) can generate content in a computationally efficient way with guaranteed performance. To solve this problem, a lightweight, gradient-free intervention that can dynamically steer trajectories away from regions corresponding to undesired meanings is proposed. The proposed method leverages the classical control theory to perform precise control on activations in a context-dependent way. In summary, this paper proposes a theoretically guaranteed generation method based on the techniques from the classic control theory.

**Audience:**

Yes

**Audience Explanation:**

Steering the behavior of LLMs has been a popular research direction in the community. This paper tries to study the problem from the perspective of control theory, which provides an interesting and new insight for the community. Thus, I think that the researcher in this field will be interested in this paper.

**Broader Impact Concerns:**

There are not any concerns on the ethical implications of the work.

**Claims And Evidence:**

Yes

**Claims Explanation:**

The content is basically fine. The claims in the paper are clear and partially supported with sound details. However, the technique details of this submission are not clear enough, and there are some drawbacks regarding the modules adopted in the paper. Further revision is required to enhance the quality of the paper.

**Requested Changes:**

- According to the content in the Intro Section, the main goal of this paper mainly focuses on "control" the generation. However, in Section3, the authors state that "In this work, we consider how to steer ...". So, since the authors claim the difference between steering and control, what is the true goal?

- How do you define the region for the desirable content? Specifically, how do you determine the $\alpha^{\rm min}$ and $\alpha^{\rm max}$?

- Since a probe classifier should be trained to map the activations into a score, how do you curate the training data for the classifier? Which model did you select as the classifier?

- Since the way of adjustment is to perform $\tilde{x}=x+\theta(x)$, assume that the input belongs to some set $x\in \mathcal{X}$, how do you ensure that $\tilde{x}\in\mathcal{X}$? In other words, how do you ensure that the modified results remain meaningful in the current semantic space?

- The mathematical formulations in this paper, especially for the controller, are rough. They should be further polished.

---

> ### Author Response · Authors · 2026-03-20
>
> We thank the reviewer for raising important points, as well as for the positive comments. Below, we explain how we have addressed the required changes:
>
> > According to the content in the Intro Section, the main goal of this paper mainly focuses on "control" the generation. However, in Section3, the authors state that "In this work, we consider how to steer ...". So, since the authors claim the difference between steering and control, what is the true goal?
>
> Thank you for pointing out this ambiguity. We agree that the wording in Section 3 was imprecise and could obscure the intended distinction between steering and control. We have revised the text to clarify that the primary goal of the paper is control at the level of latent representations, not merely steering. Specifically, our method enforces hard constraints on the activations during generation, with formal guarantees that the latent trajectory remains within a prescribed region of representation space. In this sense, the intervention constitutes control in activation space. At the level of generated tokens, however, we deliberately use the term steering rather than control. While controlled activations induce reliable and fine-grained changes in output attributes, providing formal guarantees directly in token space would require additional assumptions about the decoding process and the mapping from representations to outputs. We therefore refrain from claiming control in output space. We make this distinction explicit in the revised manuscript and refer to the induced effects on generated text as steering, while reserving the term control for the representation space where guarantees hold. For completeness, we provide sufficient conditions under which activation-level control translates to output-level guarantees in Appendix C.

---

> > ### Author Response · Authors · 2026-03-20
> >
> > > How do you define the region for the desirable content? Specifically, how do you determine the $\alpha^{\min}$ and $\alpha^{\max}$?
> >
> > The desirable region is defined through user-specified bounds on an attribute score, reflecting application-level requirements (e.g., “I can tolerate toxicity ≤ 0.5 or sentiment in [0.3, 0.7]”). These bounds are not learned by the method, but are provided as part of the control specification. To operationalize this in representation space, we use a linear probe that maps latent activations to a scalar attribute score. The parameters α_min and α_max⁡​ correspond to lower and upper bounds on this score, defining a convex region of admissible activations. The controller then enforces that the projected activation remains within this interval. This formulation mirrors standard practice in control, where admissible regions are specified externally and guarantees are provided relative to those specifications. We now clarify this explicitly in the text and have added a note before Eq. (4) to emphasize that αmin⁡ and αmax⁡ are user-defined and application-dependent.

---

> > > ### Author Response · Authors · 2026-03-20
> > >
> > > > Since a probe classifier should be trained to map the activations into a score, how do you curate the training data for the classifier? Which model did you select as the classifier?
> > >
> > > We describe how we curated the training sets in Section 5.1.1. We took the Kaggle Jigsaw Toxicity dataset as an off-the-shelf training set, and scored the generations using Logacheva et al. 2022’s toxicity scorer (see Attribute scoring functions). We compiled the sentiment dataset ourselves from a variety of domains: IMDb, tweets, Yelp reviews, and Amazon reviews. Further dataset details for sentiment were given in Appendix G Data preprocessing. Linear probes are a binary logistic regression. Implementation details, including hyperparameters, were given in Section 5.1.2 and Appendix H. The specific models used as classifiers were cited in the main and linked in Appendix B.

---

> > > > ### Author Response · Authors · 2026-03-20
> > > >
> > > > > How do you ensure that the modified results remain meaningful in the current semantic space?
> > > >
> > > > This is an important question. Our approach does not explicitly enforce membership in a semantic manifold; instead, it relies on two complementary properties that together preserve meaningfulness. First, the admissible region (desirable and undesirable sets) is defined using representations obtained from naturally generated text, via linear probes trained on meaningful examples. As a result, the constraint set lies within the region of activation space that the model already visits during normal generation. Second, the intervention is formulated as a minimum-norm correction that projects the current activation onto the admissible region whenever a constraint violation occurs. Among all interventions that satisfy the constraint, our controller selects the one with the smallest magnitude in activation space. This ensures that the modified activation remains as close as possible to the original trajectory of the model. Intuitively, this minimality property prevents the intervention from introducing large or unnatural deviations that could move the activation into regions of representation space that are not associated with coherent language. If the admissible region intersects the manifold of naturally occurring activations, the projection remains on (or arbitrarily close to) that manifold; if not, the controller still produces the smallest possible deviation required to satisfy the constraint. We now clarify this intuition in the manuscript (page 7) and explicitly connect the minimality of the control to preservation of semantic coherence.
> > > >
> > > > > The mathematical formulations in this paper, especially for the controller, are rough. They should be further polished.
> > > >
> > > > Thank you for the comment! We have revised equations (7) and (8) to introduce a more rigorous mathematical expression for the optimal control problem.

---

### Review · Reviewer_d7S7 · 2026-03-06

**Summary Of Contributions:**

This work examines the problem of steering LLM latent activations so that the final output tokens fall within a given range of an evaluator model that takes input text and produces a continuous score in the range [0, 1]. In practice the score represents a text trait like text toxicity. The method has two parts. First, a pretrained model is selected and hidden tokens are produced for many input prompts, along with a score of the toxicity of the generated content. For each hidden later, a linear probe is learned using logistic regression that take the hidden latent as input and aims to match the evaluator score. The linear probes are used to steer activation during generation in the second part. During the generation of each token, sequentially in each model layer, the hidden activations are perturbed by a closed-form value that depends on the desired evaluator range, the current hidden activation, and the linear probe weight to produce a new hidden activation that will fall in the desired range. Since this is enforced at each layer, the final generated visible token should also contain text that falls in the desired range of the evaluator. Experiments show that this method can target precise ranges of off-the-shelf evaluators for sentiment and toxicity while maintaining good text quality as measured by perplexity.

**Audience:**

Yes

**Audience Explanation:**

Yes, this method is very relevant in the context of steering and controlling LLM outputs to meet certain properties (for text safety, for example).

**Broader Impact Concerns:**

The broader impacts involve the ability to steer LLM outputs. In this work, the safety of the model could be both increased or decreased, which could lead to positive or negative uses depending on the user.

**Claims And Evidence:**

Yes

**Claims Explanation:**

The method is clearly presented and makes intuitive sense. Experiments are presented to show that linear probes can classify hidden activations into evaluator ranges with high accuracy, and that precise areas of different evaluators can be targeted while maintaining low text perplexity.

**Requested Changes:**

* In my view, the guarantees provided by this model are somewhat overstated. It heavily relies on the assumptions that 1) linear probes can accurately calibrate hidden activations with evaluator scores and 2) the linear probes are generalize to any input. The authors explore 1), but do not present thorough analysis of the second concern. If the data used to train linear probes falls in a significantly distribution compared to the test data, the guarantee does not hold. It would be great if the authors could investigate the effect of training the linear probes on significantly different text distributions compared to the test text to evaluate the robustness of the probing method. In general I feel like the word guarantee is too strong and that the claim should be softened.

* I am a little confused how text properties can be maintained on a per-token level. If the input prompt is very toxic, it does not seem reasonable that the first few tokens generated by the model can be both non-toxic and compatible with the prompt. Can you elaborate more on the relation between the length/toxicity of the prompt and amount of generation needed to meet the evaluator requirements?

* Perplexity is a useful metric, but it would be helpful to have other benchmarks like instruction following to see what kind of model capabilities are maintained. This is not a major request but important for future work.

---

> ### Author Response · Authors · 2026-03-21
>
> We thank the reviewer for the thorough comments and suggestions! **An important update is that we have added another use-case English->Spanish steering to the experiments.** Below, we address the points raised:
>
> > The guarantees provided by this model are somewhat overstated. It heavily relies on the assumptions that 1) linear probes can accurately calibrate hidden activations with evaluator scores and 2) the linear probes are generalize to any input
>
> We thank the reviewer for this thoughtful point. We agree that the notion of “guarantees” can be misinterpreted if not carefully qualified. In our framework, the guarantee strictly applies to the optimization problem: namely, that the activation is placed in a region satisfying the imposed constraint. However, the connection between controlled activations and resulting output properties is supported only by empirical evidence, which is ultimately the quantity of interest. As the reviewer correctly notes, this connection depends critically on the learned linear probe, and therefore on the data distribution used during its training. While we provide some analysis of this relationship (e.g., Lemma C.1), we acknowledge that the generalization of probes under distribution shift is not yet fully characterized. To address this, we will (i) add experiments evaluating probe robustness under distributional mismatch (e.g., training and testing on differing text domains), and (ii) revise the wording throughout the paper to soften claims of “guarantees,” instead emphasizing that these are theoretical properties of the optimization objective rather than assurances about output behavior.
>
> We have provided new experiments on OOD data (poems, parliamentary proceedings, and code). In all but one model (Gemma), LiSeCo generalizes very well to OOD datasets (see new Table 2). We emphasized that guarantees hold *in-distribution* and highlighted OOD generalization as an important direction for future work in the limitations.
>
> > Can you elaborate more on the relation between the length/toxicity of the prompt and amount of generation needed to meet the evaluator requirements?
>
> Regarding per-token control, we agree that certain properties—such as toxicity or language—may require multiple tokens to fully manifest or be corrected (cf Wolf et al., 2024). Our method does not guarantee that the very first generated token satisfies the target property; rather, it steers the trajectory of outputs so that the desired property emerges over subsequent tokens. We have added clarifications in the Limitations section to make this explicit. Qualitatively, a short prefix of generated tokens was often required before the intended behavior becomes apparent. For instance, when steering English -> Spanish, the Spanish would kick in after one or two words, often at the beginning of a clause. This is now added to the Limitations as an important direction for future work.
>
> > Perplexity is a useful metric, but it would be helpful to have other benchmarks like instruction following to see what kind of model capabilities are maintained. This is not a major request but important for future work.
>
> We thank the reviewer for this helpful suggestion. We agree that perplexity alone does not fully capture the preservation of model capabilities. In response, we have expanded the Limitations section to explicitly acknowledge this gap.

---

### Review · Reviewer_qL7x · 2026-03-16

**Summary Of Contributions:**

This paper addresses the issue of post-hoc controlling the generation of a trained language model, making a formal distinction between steering and control, with the latter entailing hard constraints and formal guarantees. The proposed method leverages optimal control theory at the level of activations, framing control as a constrained optimization problem. For each activation layer (or a subset thereof), a linear probe is trained to predict the score, which is then used during generation to constrain activations to a safe region. The general problem is formulated as a constrained trajectory optimization problem with nonlinear dynamics and constraints, which is subsequently relaxed to a locally optimal solution, solvable via a standard closed-form expression. The approach is demonstrated with examples controlling for toxicity and sentiment.

Strengths:
- In general, the paper is very clear. Details are presented in sufficient detail, and intuition about the proposed methods are provided in great detail, which makes it very easy to follow.
- The proposed method is clear and simple, well motivated, and appears to work well.

Weaknesses
- Experimental validation is fairly limited. Additional experiments or ablations could give more insight into the method: For example to examine which layers are important for constraint satisfaction and naturalness, and a more detailed analysis of the importance of the quality of the linear probes.

**Additional Comments:**

"In this case, scores α are the likelihood a sentence has a certain attribute." Could it also be the degree to which a sentence has this attribute?

"Note that the learning task is regression-like, not classification-like– we want the probes to be calibrated to the scoring
function, not just the binary labels." I assume much of this would also work with binary scores?

It was not clear to me from Section 4.1 if you train on full scored sequences or on subsequences. Would it not be beneficial to train on scored subsequences, since, to my understanding, the probes will be used in an autoregressive setting at generation time?

The performance guarantees hinge on the linear probes being precise. You comment only a little on this in Remark 4.3 and Section 5.1.2, as well as the conclusion. Could this be formalized?

"The optimization problem is 8 is relaxed into" Maybe this should be formulated as "reduced to" rather than "relaxed into".

A few thoughts:
- Generalization to more expressive non-linear probes could be an interesting direction, as well as the satisfaction of multiple constraints—both should be fairly direct extensions of the proposed method.
- The proposed method appears to be complementary to instruction tuning and prompt engineering. Would it be possible to combine the methods, and does LiSeCo provide additional benefit?
- It could be interesting to look more into which layers are most important. For example, the different layers might make different tradeoffs between constraints and naturalness. It is not clear to me that the best strategy is to use the last layers.

**Audience:**

Yes

**Audience Explanation:**

Control / steering of generative models is of broad interest, and I belive this paper makes a significant contribution.

**Broader Impact Concerns:**

No concerns

**Claims And Evidence:**

Yes

**Claims Explanation:**

I noticed a few claims about novelty and performance that may need to be adjusted.

"This is enabled by the use of an optimal control framework to cast the problem for the first time in a domain that has been overwhelmingly empirical." I am not familiar enough with the literature to know if this claim is accurate.

"This affords control over generation at a level of granularity and transparency unmatched by existing approaches." Is this claim accurate, or should it be reduced?

**Requested Changes:**

No changes requested: I recommend publication as is.

I must note that I am not an expert in the field of language models, making it difficult for me to assess whether the novelty claims are substantiated, and it is also unclear if the tasks selected to demonstrate the algorithm are sufficient to evaluate the merits of the proposed method.

---

> ### Author Response · Authors · 2026-03-20
>
> We thank the reviewer for the thorough comments and suggestions! Below, we address the points raised:
>
> > "This is enabled by the use of an optimal control framework to cast the problem for the first time in a domain that has been overwhelmingly empirical." I am not familiar enough with the literature to know if this claim is accurate.
>
> We have softened the claims that argue for novelty of the approach. While we believe that the claims are still true, this is a rapidly changing field and it is possible that some newly developed approaches share some commonalities with ours. As such, we have modified the phrasing highlighted by the reviewer.
>
> > "In this case, scores α are the likelihood a sentence has a certain attribute." Could it also be the degree to which a sentence has this attribute?
>
> Yes. The scores $\alpha$ can be interpreted both as either probabilities as well as the degree to which a feature is present. In that case, a sigmoid would not be necessary (although it can still be used) and other mechanisms such as a Likert scale could be used as well.
>
> > "Note that the learning task is regression-like, not classification-like– we want the probes to be calibrated to the scoring function, not just the binary labels." I assume much of this would also work with binary scores?
>
> Exactly, the techniques described in this paper can be used for binary labels as well. *We have added new experiments* (English -> Spanish steering) that illustrate this using binary labels (LiSeCo works quite well in this setting). Binary labels would make the presented method more similar to existing methods in the literature. One of LiSeCo's advantages in phrasing the problem as an optimal control problem with explicit bounds is that we can take advantage of exploring the full range of the feature score, which empirically leads to better tunability of the output as shown in Fig. 6.
>
> > It was not clear to me from Section 4.1 if you train on full scored sequences or on subsequences. Would it not be beneficial to train on scored subsequences, since, to my understanding, the probes will be used in an autoregressive setting at generation time?
>
> This is a great point. Currently, steering in the literature tends to focus on complete sequences (we are not aware of work on subsequences). We had actually tried extending LiSeCo to “look-ahead”, i.e., the linear probes predict future toxicity on subsequences. Empirically this performed worse than using complete sequences. We think that it’s due to the probes overfitting, due to poor data coverage of prefixes. As an example, consider the string “The person…”, which prefixes a toxic sequence in the training data. If this prefix only occurs once in the dataset, then the probes would “overfit” by always classifying sentences that start similarly as toxic.
>
> > The performance guarantees hinge on the linear probes being precise. You comment only a little on this in Remark 4.3 and Section 5.1.2, as well as the conclusion. Could this be formalized?
>
> Thanks for this point. The learned hyperplane by LiSeCo only approximates the “true” one. This approximation error might interact with the alpha range set by the practitioner. One way to formalize this might be to quantify the approximation error on the true hyperplane, and propagate this error to the guarantee. This would require more theoretical work, and investigating this further is a priority for future work.
>
> > Generalization to more expressive non-linear probes could be an interesting direction, as well as the satisfaction of multiple constraints—both should be fairly direct extensions of the proposed method.
>
> Thank you for the suggestions! Using a nonlinear probe would generalize the method. However, one of the goals of this paper was to present an intervention that is fast to compute. This is possible with a linear classifier since the optimization has a closed form. It is also possible with some certain classes of nonlinearities, and we would like to explore this in future work to make our control approach more expressive.
>
> > The proposed method appears to be complementary to instruction tuning and prompt engineering. Would it be possible to combine the methods, and does LiSeCo provide additional benefit?
>
> We believe combining this method with instruction tuning would be powerful, as a good prompt provides a good initial condition that may require less control intervention to steer into the desired feature range. We have not explored this and defer further investigations to future work.
>
> > It could be interesting to look more into which layers are most important.
>
> We definitely agree. Ablations are now added to Appendix M. Qualitatively, the results from Appendix M generalized to all tasks and models, where in general multiple-layer steering was better than single-layer steering, but steering the first layers of the model tended to degrade performance (usually PPL).

---

> > ### Comment · Reviewer_qL7x · 2026-03-23
> >
> > Thank you for the detailed response. After reading the other reviews and authors' response, I have no further reservations.

---

### Author Response · Authors · 2026-03-20

Thanks to all reviewers for the time you spent reviewing our paper. We have adjusted it accordingly and believe it is much better with your feedback. Please note the **new experiments**, where

(1) LiSeCo successfully steers English$\to$Spanish;

(2) A stress-test on out-of-distribution generalization to different datasets (code, poems, parliamentary proceedings) where LiSeCo performed very well OOD, but less so for Gemma.

(3) We have added example generations for English to Spanish in Table N.1 at the end of the Appendix.

(4) We have added ablations on the LiSeCo layer to show that steering the last layers of the model is a reasonable choice.

---

### Decision · Action_Editor_Qtdt · 2026-04-26

**Recommendation:** Accept as is

**Additional Comments:**

This paper addresses the issue of post-hoc controlling the generation of a trained language model, making a formal distinction between steering and control, with the latter entailing hard constraints and formal guarantees. The proposed method leverages optimal control theory at the level of activations, framing control as a constrained optimization problem. For each activation layer (or a subset thereof), a linear probe is trained to predict the score, which is then used during generation to constrain activations to a safe region. The general problem is formulated as a constrained trajectory optimization problem with nonlinear dynamics and constraints, which is subsequently relaxed to a locally optimal solution, solvable via a standard closed-form expression. The approach is demonstrated with examples controlling for toxicity and sentiment. In general, the paper is very clear. Details are presented in sufficient detail, and intuition about the proposed methods are provided in great detail. The proposed method is clear and simple, well motivated, and appears to work well.

**Audience:**

Yes

**Audience Explanation:**

Many researchers in controllable generation, interpretability, and model steering are actively exploring linear methods, and a well-structured taxonomy could be valuable. The manuscript’s ambition to organize emerging practices and provide conceptual scaffolding aligns with the interests of the TMLR community. With stronger evidence and clearer differentiation from prior surveys, the work could become a useful reference point.

**Claims And Evidence:**

Yes

**Claims Explanation:**

The claims made in the paper are supported by evidence. The method is clearly presented and makes intuitive sense. Experiments are presented to show that linear probes can classify hidden activations into evaluator ranges with high accuracy, and that precise areas of different evaluators can be targeted while maintaining low text perplexity.